# MEDGUARDS: MULTI-AGENT SYSTEM FOR RELIABLE MEDICAL ERROR DETECTION AND CORRECTION

## ABSTRACT

As Large Language Models (LLMs) are increasingly deployed in healthcare settings, accurate error detection and correction in generated or existing text becomes critical, as even minor mistakes can pose risks to patient safety. Existing methods for error detection and correction, including automated checks and heuristic-based approaches, do not generalize well across unseen datasets. In this paper, we propose `MedGuards` as a medical safety guardrail, which is a new framework that treats medical error detection and correction as a multi-agent in-context learning task. Specialized agents separately detect, localize, and correct errors, while a confidence-guided arbitration mechanism resolves disagreements using reasoning traces and confidence scores. This design enhances interpretability, robustness, and adaptability, without requiring additional training of the base LLMs. Additionally, we introduce the Keyword-Prioritized Correction Score (KPCS), a new evaluation metric that considers whether critical keywords within the reference text are generated correctly, providing a more comprehensive assessment than conventional metrics. Experiments across four multilingual medical datasets consisting of clinical notes demonstrate significant improvements by the proposed framework across several metrics and models. Our aim is to enable safer deployment of LLMs in real-world healthcare applications. For reproducibility, we make our code publicly available at `https://anonymous.4open.science/r/MedGuards-52F2/`.

## 1 INTRODUCTION

Large Language Models (LLMs) have demonstrated remarkable potential in various medical domains, including clinical documentation (Han et al., 2024; Williams et al., 2025), diagnostic assistance (Yang et al., 2024a; Liu et al., 2025), and personalized treatment recommendations (Yang et al., 2024b; Aththanagoda et al., 2025). Their capacity to generate coherent and contextually relevant medical text has attracted substantial interest for aiding healthcare delivery (Yang et al., 2022; Thirunavukarasu et al., 2023; Naveed et al., 2025). Existing applications of LLMs often lack dedicated error detection and correction mechanisms, a gap that is particularly critical in high-stakes clinical settings (Abacha et al., 2025). Ensuring the clinical correctness and safety of existing or generated content remains an urgent and unresolved challenge. Current quality control methods focus on assessing outputs on the semantic level, which is insufficient for addressing errors that require deep structural understanding and clinical reasoning (de Hond et al., 2024; Strong et al., 2025).

To tackle the aforementioned problems, we frame medical error detection and correction as a multi-agent in-context learning problem. Our motivation is two-fold. First, the error correction process involves step-wise reasoning comprising detection, localization, and correction. This structure aligns well with the scenarios paradigm and Chain-of-Thought (CoT) reasoning (Wei et al., 2022), both of which decompose complex reasoning into interpretable, context-aware steps. Second, medical applications demand high reliability, which may be difficult to ensure when relying on a single model subject to uncertainty, bias, or failure in out-of-distribution scenarios (Hong et al., 2024; Atf et al., 2025). Inspired by self-consistency (Wang et al., 2023), we propose multiple specialized agents that compute predictions separately and then aggregate their outputs where necessary, thereby reducing prediction variance and improving robustness. We also leverage In-Context Learning (ICL) (Xu et al., 2024), allowing agents to incorporate the reasoning traces and confidence scores of peer agents directly into their prompts. By combining these three principles, CoT for task decomposi-

tion, self-consistency for performance robustness, and ICL for adaptive decision-making, we design a principled framework that is both interpretable and practically effective for medical error detection and correction.

In terms of evaluation, existing metrics for natural language processing applications, such as BLEU (Papineni et al., 2002), ROUGE (Lin, 2004), and BERTScore (Zhang et al., 2020), treat each token equally when computing similarity between the generated and reference text. However, this token-level equality assumption introduces a critical limitation: these metrics fail to account for the varying importance of different tokens, especially those representing key clinical entities within the reference text. As a result, errors in generating essential terms may be overlooked despite high overall similarity scores. For example, a model may generate a suspected causal organism based on the patient's symptoms as: "*Patient's symptoms are suspected to be due to hepatitis A*", compared to the reference text of "*Patient's symptoms are suspected to be due to Schistosoma mansoni*". Although the surrounding sentence remains largely unchanged, the diagnostic entity itself is fundamentally different. Such clinically critical entities, e.g., causal organism or diagnoses, play a decisive role in determining whether a correction is clinically meaningful. Therefore, treating each token as equally important may be insufficient to reliably evaluate the quality of generated clinical notes.

In this paper, we propose a systematic approach to error detection and correction, aiming to enhance reliability and facilitate safer deployment of LLMs in clinical environments. We also propose a novel evaluation metric, which emphasizes the identification and validation of critical clinical entities in the reference text. Our main contributions can be summarized as follows:

- We design `MedGuards`, the first multi-agent system tailored for medical error detection and correction. `MedGuards` consists of three collaborative modules: detection, localization, and correction. Each agent specializes in a sub-task, and the system is plug-and-play and highly flexible, making it easy to integrate with existing LLM-based medical agents without additional re-training. This design also provides fine-grained interpretability and supports scalable deployment across diverse medical domains.

- We propose a confidence-guided ICL framework to reach consensus when disagreements arise. Specifically, when two agents output conflicting predictions, their reasoning steps and confidence scores are passed to a third decision agent, which adjudicates the conflict. This improves robustness in uncertain scenarios, enabling more reliable error correction.

- We introduce a new evaluation metric that prioritizes domain-critical medical keywords. Unlike traditional metrics that treat all tokens equally, the proposed metric checks whether critical keywords within the reference text are generated correctly and assesses the overall semantics. This balanced design better reflects the clinical safety and reliability requirements of medical error correction.

- We extensively evaluate our system on four datasets, covering three languages (English, Arabic, and Chinese) and show consistent improvements over strong baselines, demonstrating the method's generalizability and robustness across diverse medical settings.

## 2 RELATED WORK

### 2.1 MEDICAL ERROR DETECTION, CORRECTION, AND EVALUATION

The release of the MEDIQA-CORR 2024 Shared Task (Abacha et al., 2024) provided the first benchmark for medical error detection and correction. Submitted systems included prompting-based LLMs with few-shot ICL and CoT reasoning (Wu et al., 2024b; Rajwal et al., 2024) and retrieval-augmented pipelines grounding corrections in external knowledge (Lewis et al., 2020). One study proposed a hybrid framework combining classifiers with Question-Answering (QA)-style modules (Saeed, 2024), while in another study the authors leveraged multiple expert prompts and self-consistency to improve robustness (Wang et al., 2022).

Evaluation in this shared task relied on BLEU (Papineni et al., 2002), ROUGE (Lin, 2004), and embedding-based metrics, such as BERTScore (Zhang et al., 2020). However, these metrics show weak alignment with expert judgments when factual accuracy and domain-specific terminology are critical (Novikova et al., 2017; Maynez et al., 2020; Fabbri et al., 2021). Domain-adapted metrics address this gap. CUI F-score measures overlap in UMLS concepts (Gao et al., 2022), SapBERTScore

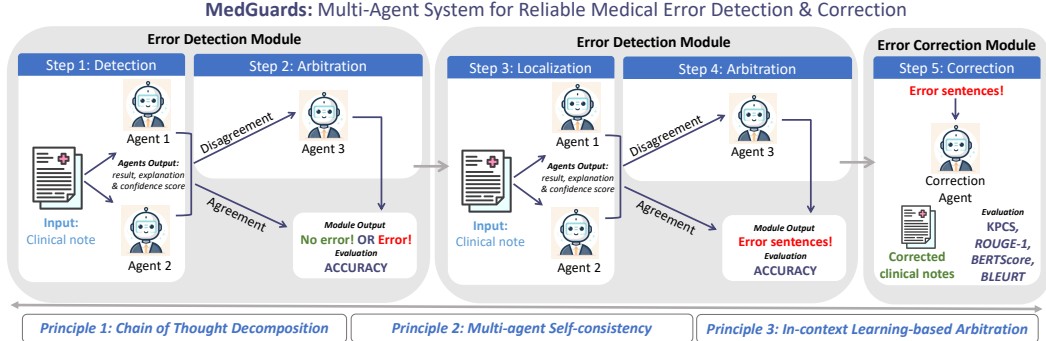

Figure 1: High-level Overview of `MedGuards`.

leverages biomedical embeddings to capture terminology similarity (Liu et al., 2020), and Clinical-BLEURT adapts BLEURT for clinical summarization and dialogue (Croxford et al., 2024). Overall, these metrics were developed for broader clinical NLP rather than error detection and correction, and no standardized framework yet exists to prioritize domain-critical entities.

## 2.2 MEDICAL MULTI-AGENT SYSTEMS

Multi-agent systems have been studied as a way to improve reasoning, reliability, and interpretability of LLMs. General frameworks decompose tasks, enable debate, or use arbitration to aggregate reasoning traces (Wu et al., 2024a; Li et al., 2023; Yao et al., 2023; Wang et al., 2022). Recent work adapts these ideas to clinical settings, by designing agentic teams for diagnosis and decision support (Yue et al., 2024; Kim et al., 2024), clinical simulation and evaluation (Schmidgall et al., 2024), rare-disease or knowledge-graph–aware differential diagnosis (Zuo et al., 2025; Chen et al., 2024), and radiology report generation with radiologist-in-the-loop agents (Yi et al., 2025). These approaches highlight the promising prospects of multi-agent coordination in clinical NLP, but none have been designed specifically for medical error detection and correction. The systematic use of arbitration to identify, localize, and correct errors remains underexplored.

## 3 METHODOLOGY

We frame medical error detection and correction as a multi-agent reasoning problem, where multiple specialized agents collaborate to analyze, debate, and refine solutions. Rather than presenting our system merely as a sequential pipeline, we ground it in three core principles that we believe are central to our multi-agent system using LLMs:

- ***Principle 1:*** CoT decomposition more explicitly articulates intermediate reasoning steps, mimicking clinical reasoning processes.

- ***Principle 2:*** Multi-agent self-consistency leverages multiple agents whose outputs can be compared, debated, thereby reducing the brittleness of relying on a single agent prediction.

- ***Principle 3:*** ICL-based arbitration enables agents to condition their judgments on the reasoning traces and confidence scores of other agents, fostering more robust decision-making.

We hypothesize that the three principles enable meta-level reasoning without additional model training. By combining these principles, we cast medical error correction not only as a practical system but also as an instantiation of generalizable reasoning paradigms in LLM research. The following sections detail the methodological details, and the complete algorithm is shown in Algorithm 1.

## 3.1 COT-GUIDED TASK DECOMPOSITION

Following the CoT paradigm, we decompose the overall task into three reasoning stages.

**Algorithm 1:** Overview of `MedGuards` framework consisting of three steps for error detection ($f^{\text{Det}}$), localization ($f^{\text{Loc}}$), and correction ($f^{\text{Corr}}$), where $r$ and $c$ indicates a reasoning trace and confidence score, respectively, without loss of generality, and $M$ is the number of matching characters among the longest common subsequences.

---

**Input:** Clinical note $x$
**Output:** Corrected sentence set $\hat{s}$ if error exists
Initialize error detection agents $\text{Agent}_i^{\text{Det}}$ where $i \in \{1, 2\}$
1. `Error Detection Step`$(f^{\text{Det}})$                    ▷ Agents process input text

$$(d_i, r_i^{\text{Det}}, c_i^{\text{Det}}) \leftarrow \text{Agent}_i^{\text{Det}}(x), \quad i \in \{1, 2\}$$

If $d_1 == d_2$:
$$d \leftarrow d_1$$

Else, initialize & invoke arbitration agent:
$$d \leftarrow \text{Agent}_3^{\text{Det}}\big(x \,\|\, (d_1, r_1^{\text{Det}}, c_1^{\text{Det}}), (d_2, r_2^{\text{Det}}, c_2^{\text{Det}})\big)$$

2. `Error Localization Step`$(f^{\text{Loc}})$                    ▷ Initialize if $d = 1$

$$(s_i, r_i^{\text{Loc}}, c_i^{\text{Loc}}) \leftarrow \text{Agent}_i^{\text{Loc}}(x), \quad i \in \{1, 2\}$$

If $s_1 == s_2$:
$$s \leftarrow s_1$$

Else, initialize & invoke arbitration agent:
$$s \leftarrow \text{Agent}_3^{\text{Loc}}\big(x \,\|\, (s_1, r_1^{\text{Loc}}, c_1^{\text{Loc}}), (s_2, r_2^{\text{Loc}}, c_2^{\text{Loc}})\big)$$

Align predicted erroneous sentence $s$ to original sentence in $x$ using string similarity:
$$s^* = \arg\max_{x_i \subset x} \text{sim}(x_i, s), \quad \text{sim}(x_i, s) = \frac{2 \cdot M}{|x_i| + |s|}.$$

3. `Error Correction Step`$(f^{\text{Corr}})$                    ▷ Generate corrected sentences

$$\hat{s} \leftarrow \text{Agent}^{\text{Corr}}(s^* \mid x)$$

Return $\hat{s}$

---

**Error Detection.** In the first step, denoted as $f^{\text{Det}}$, multiple agents collaborate to detect whether an error exists in the input clinical note $x$, such that $d = f^{\text{Det}}$, where $d \in \{0, 1\}$ indicates the presence of an error.

**Error Localization.** In the second step, the goal of the multi-agent system is to identify the sentences $s \subset x$ containing the error with a multi-agent error localization function $f^{\text{Loc}}$, such that $s = f^{\text{Loc}}(x \mid d)$.

**Error Correction.** In the final step, an agent is expected to generate a corrected version $\hat{s}$ of the erroneous sentences. Once the erroneous sentence set $s$ is localized, a dedicated correction agent, with the function $f^{\text{Corr}}$, generates the corrected version $\hat{s}$, such that $\hat{s} = f^{\text{Corr}}(s \mid x)$. Formally:

$$d = f^{\text{Det}}(x), \quad s = f^{\text{Loc}}(x \mid d), \quad \hat{s} = f^{\text{Corr}}(s \mid x). \tag{1}$$

This decomposition aligns task structure with CoT reasoning, ensuring explicit intermediate steps.

### 3.2 MULTI-AGENT SELF-CONSISTENCY

To enhance reliability under uncertainty, we propose a *multi-agent self-consistency* framework. Rather than committing to the output of a single model, multiple agents separately reason over the same input, and their outputs are brought to consensus through structured arbitration. This design extends the classical notion of self-consistency (Wang et al., 2023) from single-model sampling to a multi-agent setting, where disagreement itself becomes a signal to trigger higher-order deliberation. We propose this framework for two complementary tasks: error detection and error localization.

**Error Detection.** Two base agents, $\text{Agent}_1^{\text{Det}}$ and $\text{Agent}_2^{\text{Det}}$, separately analyze the clinical text $x$ and produce binary predictions $d_1, d_2 \in \{0, 1\}$, accompanied by reasoning traces $r_1^{\text{Det}}, r_2^{\text{Det}}$ and

confidence scores $c_1^{\text{Det}}, c_2^{\text{Det}}$:

$$(d_i, r_i^{\text{Det}}, c_i^{\text{Det}}) = \text{Agent}_i^{\text{Det}}(x), \quad i \in \{1, 2\}, \ c_i^{\text{Det}} \in [0, 1]. \tag{2}$$

Agreement between $d_1$ and $d_2$ yields a direct consensus. In case of disagreement, an arbitration agent $\text{Agent}_3^{\text{Det}}$ is activated to reach consensus, thereby converting the binary conflict into a resolved decision. The arbitration agent will consider the original inputs, candidate sets, reasoning traces, and confidence scores for a final decision.

**Error Localization.** For localization, each agent $\text{Agent}_i^{\text{Loc}}$ examines the full input text $x$ and outputs a candidate erroneous sentence $s_i \subseteq x$ together with a reasoning trace $r_i^{\text{Loc}}$ and a confidence score $c_i^{\text{Loc}}$:

$$(s_i, r_i^{\text{Loc}}, c_i^{\text{Loc}}) = \text{Agent}_i^{\text{Loc}}(x), \quad i \in \{1, 2\}, \ c_i^{\text{Loc}} \in [0, 1]. \tag{3}$$

Consensus is immediately reached if $s_1 = s_2$, otherwise, an arbitration agent $\text{Agent}_3^{\text{Loc}}$ consolidates the original inputs, candidate sets, reasoning traces, and confidence scores for a final decision. This process generalizes the principle of self-consistency: detection is resolved through binary consensus with arbitration, whereas localization requires deliberative reconciliation of structured outputs. In both cases, multi-agent disagreement is not treated as noise, but rather enables more reliable inference. To automatically align an erroneous sentence with a correction candidate, we use the SequenceMatcher (Python Software Foundation, 2025) to compute character-level similarity between the generated sentence $x_i$ and each sentence in the reference clinical notes, selecting the sentence $s$ with the highest score based on the computed similarity ratio. We adopt character-level similarity because localization aims to identify the exact sentence from the reference text, not a paraphrased or semantically similar one. This could be possible within a long medical note and LLMs may occasionally produce minor lexical variations or hallucinations.

The similarity ratio is defined as:

$$\text{sim}(x_i, s) = \frac{2 \cdot M}{|x_i| + |s|}, \tag{4}$$

where $M$ is the number of matching characters in the longest common subsequences between $x_i$ and $y$. The resulting score lies in [0,1], with higher values indicating greater string-level similarity. This metric allows us to identify the sentence in the report that is most similar to the candidate erroneous sentence, without relying on any semantic representations.

## 3.3 ICL-BASED PREDICTION

We formulate arbitration as an ICL problem. The arbitration agents do not rely on additional parameter updates but instead condition their reasoning on the entire contexts, including original inputs, candidate sets, reasoning traces, and confidence scores of other agents provided in the prompt.

**Error Detection.** The arbitration agent $\text{Agent}_3^{\text{Det}}$ receives as context the reasoning traces and confidence scores from $\text{Agent}_1^{\text{Det}}$ and $\text{Agent}_2^{\text{Det}}$:

$$d = \text{Agent}_3^{\text{Det}}\big(x \parallel (d_1, r_1^{\text{Det}}, c_1^{\text{Det}}), (d_2, r_2^{\text{Det}}, c_2^{\text{Det}})\big), \tag{5}$$

where $\parallel$ denotes concatenation into the input prompt. In this way, the arbitrator leverages prior reasoning as demonstrations to guide its final decision.

**Error Localization.** Similarly, $\text{Agent}_3^{\text{Loc}}$ is prompted with the candidate sentences and rationales from $\text{Agent}_1^{\text{Loc}}$ and $\text{Agent}_2^{\text{Loc}}$:

$$s = \text{Agent}_3^{\text{Loc}}\Big(x \parallel (s_1, r_1^{\text{Loc}}, c_1^{\text{Loc}}), (s_2, r_2^{\text{Loc}}, c_2^{\text{Loc}})\Big). \tag{6}$$

This setting reframes debate resolution as an ICL task, where the arbitrator agent learns to synthesize final decisions from conflicting rationales provided in-context.

## 3.4 KEYWORD-PRIORITIZED CORRECTION SCORE

To better evaluate medical error correction, we propose the Keyword-Prioritized Correction Score (KPCS), which explicitly incorporates domain-specific keywords to ensure corrections remain both fluent and clinically accurate. The keywords refer to key clinical concepts in the reference text, such as diagnoses, medications or therapeutic actions, that are central to the clinical meaning of each note

and particularly relevant to potential errors. Examples of the annotated keywords can be found in Appendix A.10. The metric is computed in three steps:

**Step 1: Keyword Check.** Given a corrected sentence $S_c$ and the reference sentence $S_r$, we first check whether $S_c$ contains the required keywords extracted from $S_r$. $n$ and $m$ denote the number of keywords in $S_c$ and $S_r$. If no keyword is present in $S_c$, the score is set to zero:

$$K(S_c, S_r) = n/m. \tag{7}$$

**Step 2: Semantic and Lexical Similarity.** We further compute the average of three widely used similarity metrics, which are ROUGE-1, BERTScore, and BLEURT:

$$M(S_c, S_r) = \frac{\text{ROUGE-1}(S_c, S_r) + \text{BERTScore}(S_c, S_r) + \text{BLEURT}(S_c, S_r)}{3}. \tag{8}$$

**Step 3: Weighted Score.** The final KPCS is a weighted combination of the keyword indicator and the similarity score:

$$\text{KPCS}(S_c, S_r) = \alpha \cdot K(S_c, S_r) + (1 - \alpha) \cdot M(S_c, S_r), \tag{9}$$

where $\alpha \in [0, 1]$ is a tunable parameter that balances the importance of keyword presence and overall semantic similarity. This flexible design enables penalizing outputs that do not include critical clinical keywords, while still rewarding overall fluency and semantic alignment.

## 4 EXPERIMENTS & RESULTS

### 4.1 DATASETS, BASELINE MODELS AND EVALUATION METRICS

We conducted experiments on the publicly available MEDEC dataset (Abacha et al., 2025) and an internal multilingual dataset consisting of English, Arabic and Chinese subsets. The details of the datasets are provided in Appendix A.1. Our baselines include a set of state-of-the-art error detection and correction models, including knowlab_AIMed (Wu et al., 2024b), EM-mixer (Rajwal et al., 2024), Medifact (Saeed, 2024), IryoNLP (Corbeil, 2024), and IKIM (Valiev & Tutubalina, 2024). We also include state-of-the-art LLMs, including Gemini 2.0 Flash, Gemini 2.5 Flash Lite, GPT-4o-mini, Doubao-1.5-thinking-pro, Deepseek-V3, and Llama-3.3-70B-Instruct. We evaluate model performance on three sub-tasks: error detection, error localization, and sentence correction. For detection and localization, we use accuracy as the main metric. For correction, we report ROUGE-1 (Lin, 2004), BERTScore (Zhang et al., 2020), BLEURT (Sellam et al., 2020), and our proposed evaluation metric KPCS. More details on experimental settings can be found in Appendix A.2.

### 4.2 OVERALL PERFORMANCE

Table 1: Comparison with existing error detection and correction methods on the MEDEC dataset. The last row reports the performance of our framework `MedGuards`, which is the best variant with Doubao-1.5 thinking pro as the base LLM. Best results per column are highlighted in bold.

| Models | Error Detection | Error Localization | Error Correction | | |
| --- | --- | --- | --- | --- | --- |
| | Accuracy | Accuracy | ROUGE-1 | BERTScore | BLEURT |
| knowlab_AIMed (Wu et al., 2024b) | 0.694 | 0.620 | 0.644 | 0.677 | 0.654 |
| EM-mixer (Rajwal et al., 2024) | 0.680 | 0.640 | 0.571 | 0.595 | 0.596 |
| Medifact (Saeed, 2024) | 0.737 | 0.600 | 0.454 | 0.444 | 0.439 |
| IryoNLP (Corbeil, 2024) | 0.671 | 0.610 | 0.561 | 0.592 | 0.591 |
| IKIM (Valiev & Tutubalina, 2024) | 0.678 | 0.590 | 0.523 | 0.564 | 0.588 |
| MedGuards | **0.770** | **0.716** | **0.724** | **0.731** | **0.695** |

Table 1 summarizes the performance of existing error detection and correction methods on the MEDEC dataset. We selected comparable error detection and correction baseline models that do not require introducing additional databases. Overall, these models show balanced but limited performance: for example, Medifact achieves the high detection accuracy (0.737) but performs worse on correction quality (ROUGE-1 = 0.454), while others like EM-mixer and IKIM remain unideal

Table 2: Evaluation of LLM backbones with and without `MedGuards` on the MEDEC dataset. Bold numbers indicate best results. Δ% shows average improvement across all metrics over the baseline.

| Models | Error Detection | Error Localization | Error Correction | | | | P-values | Δ% |
|---|---|---|---|---|---|---|---|---|
| | Accuracy | Accuracy | ROUGE-1 | BERTScore | BLEURT | KPCS | | |
| Gemini 2.0 Flash | 0.589 | 0.415 | 0.379 | 0.380 | 0.395 | 0.370 | 4.03E-04 | – |
| + MedGuards | **0.713** | **0.609** | **0.537** | **0.549** | **0.549** | **0.472** | – | +35.6% |
| GPT-4o-mini | 0.539 | 0.374 | 0.300 | 0.296 | 0.335 | 0.210 | 8.31E-05 | – |
| + MedGuards | **0.695** | **0.535** | **0.571** | **0.581** | **0.574** | **0.461** | – | +66.2% |
| Doubao-1.5-thinking-pro | 0.696 | 0.318 | 0.596 | 0.618 | 0.624 | 0.219 | 1.33E-02 | – |
| + MedGuards | **0.770** | **0.716** | **0.724** | **0.731** | **0.695** | **0.554** | – | +36.4% |
| Deepseek-V3 | 0.533 | 0.413 | 0.562 | 0.547 | 0.557 | 0.257 | 3.15E-04 | – |
| + MedGuards | **0.681** | **0.679** | **0.708** | **0.712** | **0.682** | **0.556** | – | +40.0% |
| Gemini 2.5 Flash Lite | 0.583 | 0.224 | 0.339 | 0.350 | 0.367 | 0.127 | 5.69E-03 | – |
| + MedGuards | **0.595** | **0.538** | **0.513** | **0.521** | **0.499** | **0.385** | – | +53.3% |
| Llama-3.3-70B-Instruct | 0.621 | 0.330 | 0.397 | 0.395 | 0.401 | 0.360 | 4.23E-03 | – |
| + MedGuards | **0.684** | **0.576** | **0.528** | **0.530** | **0.536** | **0.487** | – | +33.10% |

across some metrics. Our model (`MedGuards` with the base LLM as Doubao-1.5 thinking pro) outperforms the best existing methods by a large margin, highlighting the generalizability and effectiveness of our framework in medical error detection and correction.

Table 2 summarizes the performance of different LLM base models with and without our multi-agent framework `MedGuards`. Overall, the results highlight the overall effectiveness of `MedGuards`. Across all backbones, `MedGuards` delivers substantial gains over the plain LLMs, with improvements spanning error detection, localization, and generation quality. For instance, Doubao-1.5-thinking-pro with `MedGuards` achieves state-of-the-art performance, reaching 0.770 in detection and 0.716 in localization, along with strong correction metrics. `MedGuards` consistently narrows the gap between weaker backbones (e.g., Gemini 2.0 Flash and GPT-4o-mini) and stronger ones, demonstrating robustness across heterogeneous LLMs. All results indicate statistically significant improvements ($p < 0.05$), demonstrating that the proposed `MedGuards` framework reliably enhances model performance. Notably, the improvement is especially evident in KPCS, where `MedGuards` consistently raises scores greatly, for example, from 0.370 to 0.472 on Gemini-2.0-Flash, from 0.210 to 0.461 on GPT-4o-mini and from 0.219 to 0.554 on Doubao-1.5-thinking-pro, indicating that corrections not only match the reference text but also preserve clinical key-phrase fidelity. Moreover, we conduct human evaluations, further validating the effectiveness of `MedGuards`, with detailed results provided in the Appendix A.3.

Table 3 shows that integrating `MedGuards` consistently improves performance across all three languages in the second dataset. In English, `MedGuards` yields substantial gains over backbone LLMs, with improvements of over 20 points in error detection (e.g., Gemini 2.0 Flash: 0.514 → 0.755) and large boosts in generation quality. Similar trends are observed in Arabic, where `MedGuards` enhances both detection and localization accuracy, with Deepseek-v3 + `MedGuards` achieving the strongest overall performance. In Chinese, we observe similar improvements, where Deepseek-v3 + `MedGuards` achieves state-of-the-art results across all metrics, surpassing both its backbone and other models by a clear margin. These results demonstrate that `MedGuards` is model-agnostic and consistently strengthens LLMs in both safety-critical error identification and faithful correction across datasets with diverse languages.

## 4.3 Additional Analysis

### 4.3.1 Ablation study

Table 4 presents an ablation study of `MedGuards` on the MEDEC dataset with select models due to resource constraints, where we vary the ICL inputs by including reasoning and confidence signals for detection and localization steps. We observe that including detection confidence and localization reasoning consistently improves both error detection and localization across different backbones. For Gemini 2.0 Flash, localization reasoning yields gain in localization accuracy and generation quality, while interestingly, localization confidence does not help much. For GPT-4o-mini, the trend

Table 3: Evaluation of LLM backbones with and without `MedGuards` on the multilingual dataset, in English, Arabic, and Chinese. Bold numbers indicate best results. $\Delta\%$ shows average improvement across all metrics over the baseline.

| Languages | Models | Error Detection | Error Localization | Error Correction | | | $\Delta\%$ |
| | | Accuracy | Accuracy | ROUGE-1 | BERTScore | BLEURT | |
|---|---|---|---|---|---|---|---|
| English | Gemini 2.0 Flash | 0.514 | 0.168 | 0.281 | 0.294 | 0.288 | – |
| | + MedGuards | **0.755** | **0.322** | **0.651** | **0.694** | **0.613** | 96.1% |
| | GPT-4o-mini | 0.664 | 0.524 | 0.487 | 0.498 | 0.472 | – |
| | + MedGuards | **0.683** | **0.580** | **0.498** | **0.511** | **0.488** | 4.38% |
| | Deepseek-v3 | 0.529 | 0.318 | 0.330 | **0.570** | 0.381 | – |
| | + MedGuards | **0.808** | **0.414** | **0.516** | 0.499 | **0.610** | 33.7% |
| | Gemini 2.5 Flash Lite | 0.567 | 0.264 | 0.349 | 0.362 | 0.346 | – |
| | + MedGuards | **0.731** | **0.274** | **0.544** | **0.615** | **0.548** | 43.5% |
| Arabic | Gemini 2.0 Flash | 0.577 | 0.175 | 0.260 | 0.469 | 0.292 | – |
| | + MedGuards | **0.711** | **0.330** | **0.418** | **0.600** | **0.445** | 41.2% |
| | GPT-4o-mini | **0.577** | 0.175 | 0.260 | 0.469 | 0.292 | – |
| | + MedGuards | 0.563 | **0.196** | **0.363** | **0.777** | **0.388** | 28.9% |
| | Deepseek-v3 | 0.670 | 0.309 | 0.366 | 0.570 | 0.389 | – |
| | + MedGuards | **0.711** | **0.371** | **0.463** | **0.622** | **0.480** | 14.8% |
| | Gemini 2.5 Flash Lite | 0.495 | 0.268 | 0.303 | 0.432 | 0.318 | – |
| | + MedGuards | **0.722** | **0.289** | **0.367** | **0.689** | **0.380** | 34.8% |
| Chinese | Gemini 2.0 Flash | 0.705 | **0.455** | 0.569 | 0.659 | 0.577 | – |
| | + MedGuards | **0.761** | 0.415 | **0.684** | **0.712** | **0.634** | 8.16% |
| | GPT-4o-mini | 0.505 | 0.115 | 0.244 | 0.390 | 0.257 | – |
| | + MedGuards | **0.600** | **0.235** | **0.595** | **0.536** | **0.529** | 65.2% |
| | Deepseek-v3 | 0.655 | 0.390 | 0.615 | 0.578 | 0.621 | – |
| | + MedGuards | **0.815** | **0.420** | **0.748** | **0.782** | **0.708** | 21.4% |
| | Gemini 2.5 Flash Lite | 0.600 | 0.375 | 0.448 | 0.533 | 0.455 | – |
| | + MedGuards | **0.625** | **0.405** | **0.516** | **0.773** | **0.501** | 17.0% |

is complementary, such that including localization confidence boosts generation fidelity, with the complete `MedGuards` setup delivering the strongest ROUGE and BLEURT scores.

Table 4: Ablation study on our proposed `MedGuards` using the MEDEC dataset. Columns indicate whether the corresponding strategy is enabled (✔) or disabled (✘). Best results per base model are highlighted in bold.

| Base Models | Ablations | | | | Error Detection | Error Localization | Error Correction | | |
| | Detection reasoning | Detection confidence | Localization reasoning | Localization confidence | Accuracy | Accuracy | ROUGE-1 | BERTScore | BLEURT |
|---|---|---|---|---|---|---|---|---|---|
| Gemini 2.0 Flash | ✔ | ✘ | ✘ | ✘ | 0.673 | 0.493 | 0.403 | 0.419 | 0.451 |
| | ✔ | ✔ | ✘ | ✘ | 0.695 | 0.558 | 0.478 | 0.494 | 0.505 |
| | ✔ | ✔ | ✔ | ✘ | 0.682 | **0.629** | **0.557** | **0.565** | **0.558** |
| | ✔ | ✔ | ✔ | ✔ | **0.713** | 0.609 | 0.537 | 0.549 | 0.549 |
| GPT-4o-mini | ✔ | ✘ | ✘ | ✘ | 0.636 | 0.409 | 0.342 | 0.361 | 0.434 |
| | ✔ | ✔ | ✘ | ✘ | 0.678 | 0.496 | 0.419 | 0.447 | 0.480 |
| | ✔ | ✔ | ✔ | ✘ | 0.676 | **0.537** | 0.483 | 0.486 | 0.484 |
| | ✔ | ✔ | ✔ | ✔ | **0.695** | 0.535 | **0.571** | **0.581** | **0.574** |

### 4.3.2 COLLABORATION STRATEGY

Table 5 compares the impact of voting and ICL debating strategies on error detection, localization, and correction quality. Enabling the ICL debating consistently improves all metrics for both Gemini 2.0 Flash and GPT-4o-mini. Notably, GPT-4o-mini benefits most from ICL, with error detection increasing from 0.653 to 0.770 and error localization from 0.474 to 0.611. These results indicate that structured multi-agent discussion substantially enhances performance beyond simple voting.

Table 5: Comparison of voting and debating strategies for detection and localization. ✔indicates whether the corresponding strategy is enabled. Best results per base model are highlighted in bold.

| Model Name | Vote | ICL | Error Detection | Error Localization | ROUGE-1 | BERTScore | BLEURT |
|---|---|---|---|---|---|---|---|
| Gemini 2.0 Flash | ✔ | | 0.693 | 0.557 | 0.466 | 0.482 | 0.490 |
| | | ✔ | **0.713** | **0.609** | **0.537** | **0.549** | **0.549** |
| GPT-4o-mini | ✔ | | 0.653 | 0.474 | 0.407 | 0.417 | 0.447 |
| | | ✔ | **0.770** | **0.611** | **0.559** | **0.570** | **0.500** |

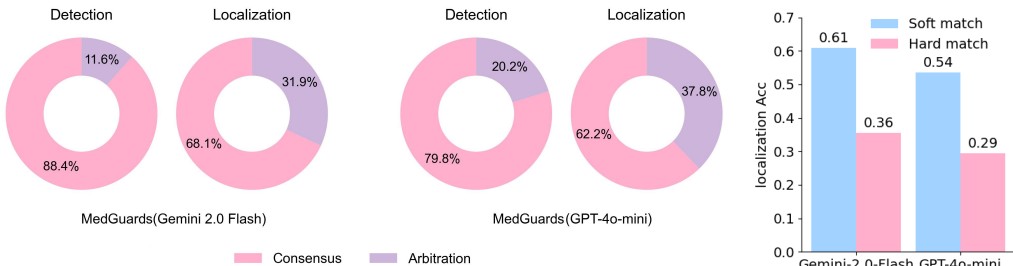

Figure 2: Distribution of self-consistency rate on the MEDEC dataset.

Figure 3: Soft vs hard match for error localization.

### 4.3.3 SELF-CONSISTENCY RATE

To better understand the role of self-consistency in `MedGuards`, we analyze the proportions of consensus versus arbitration within the self-consistency framework across different base models and stages. Specifically, we report statistics on both detection and localization steps. Figure 2 presents the results on the MEDEC dataset. Additional results on the different languages are provided in Appendix A.4. As shown in Figure 2, for detection, the majority of predictions rely on consensus, with arbitration contributing only a small fraction (11.6% for Gemini 2.0 Flash and 20.2% for GPT-4o-mini). This indicates that detection is generally stable across agents. In contrast, for localization, the reliance on arbitration increases notably (31.9% for Gemini 2.0 Flash and 37.8% for GPT-4o-mini), suggesting that error localization is inherently more ambiguous and benefits from arbitration. Moreover, GPT-4o-mini shows higher arbitration usage than Gemini 2.0 Flash in both detection and localization, reflecting model-dependent differences in self-consistency dynamics.

### 4.3.4 SOFT VS HARD MATCHING FOR ERROR LOCALIZATION

Figure 3 compares soft and hard matching strategies for error sentence localization. In the soft matching setting, the model generates a complete sentence, which is then aligned with the most similar sentence within the clinical notes. The most similar candidate is selected as the final localized sentence. In contrast, the hard matching setting requires the model to directly output the sentence index, and the prediction is considered correct only if it exactly matches the ground-truth erroneous sentence. As shown in Figure 3, soft matching achieves substantially higher localization accuracy (0.61 for Gemini 2.0 Flash and 0.54 for GPT-4o-mini) than hard matching (0.36 and 0.29). This highlights that while strict sentence-level alignment remains challenging, allowing semantic similarity in the soft matching provides a more robust evaluation of localization ability. Additionally, we provide additional results on using mixtures of agents in Appendix A.5, latency and computational requirements in Appendix A.6, different multi-agent design structures in Appendix A.7, sensitivity analysis of KPCS in Appendix A.8, and assessment of qualitative behavior of agents in Appendix A.9. Overall, the results demonstrate the robustness of our proposed framework.

## 5 CONCLUSION

In this work, we introduce `MedGuards`, the first multi-agent framework specifically designed for medical error detection and correction in LLM-generated clinical text. By decomposing the task into error detection, localization, and correction, and coordinating these specialized agents through a confidence-guided arbitration mechanism, our approach achieves higher robustness and interpretability while remaining plug-and-play for existing medical assistants. To further ensure clinical reliability, we proposed KPCS, a novel evaluation metric that explicitly prioritizes domain-critical factual entities, offering a more safety-oriented and clinically meaningful assessment compared to existing metrics. Extensive experiments across four datasets and three languages demonstrated the effectiveness and generality of our framework, consistently outperforming strong baselines. `MedGuards` offers a systematic solution towards safer deployment of medical LLMs, enabling more trustworthy integration of generative models into real-world clinical environments.

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

# A  APPENDIX

## A.1  DATASETS, BASELINE MODELS AND EVALUATION METRICS

We conducted experiments on the publicly available MEDEC dataset (Abacha et al., 2025) and an internal multilingual dataset, consisting of English, Arabic, and Chinese subsets. MEDEC is the first benchmark for medical error detection and correction in clinical notes. It contains 3,848 clinical texts each either correct or containing a single error in one of five categories: Diagnosis, Management, Treatment, Pharmacotherapy, or Causal Organism. The internal dataset was constructed to evaluate multilingual medical-error correction and contains 2,506 clinical texts across three languages: English (1,024), Chinese (1,000), and Arabic (482). The English and Chinese subsets were sampled and perturbed from MedQA (Jin et al., 2021), while the Arabic subset was derived from MedArabiQ (Daoud et al., 2025). Each item was designed to cover one of ten clinician-defined medical error categories: diagnosis, management, treatment, pharmacotherapy, causal organism/pathogen, lab/serum value interpretation, physiology, histology, anatomy, and epidemiology. To ensure quality, we applied a two-step human review process: (1) NLP researchers verified fluency and plausibility, and (2) two independent clinicians per language validated medical correctness and clinical realism. No translations were used to preserve language-specific nuances. All keywords were manually annotated and verified by clinicians across all samples. The selection procedure can also be automated by comparing each error sentence with its corrected counterpart using an LLM. We include the LLM script used for this extraction, so new datasets can be processed without manual labeling.

Our baselines include both prior systems from the MEDIQA-CORR 2024 Shared Task and a set of recent large language models. Specifically, knowlab_AIMed (Wu et al., 2024b) formulates the task using few-shot in-context learning with chain-of-thought prompting, where error detection is treated as a binary decision, localization requires extracting the erroneous sentence, and correction relies on constrained rewriting. EM-mixers (Rajwal et al., 2024) extend this paradigm by injecting entity- and knowledge-aware cues into the prompts and aggregating multiple exemplars to strengthen detection and guide correction. Medifact (Saeed, 2024) adopts a hybrid design, combining a lightweight classifier for detection with a QA-style correction module that emphasizes factual consistency and interpretability. IryoNLP (Corbeil, 2024) decomposes the task into specialized roles whose outputs are aggregated through coordination rules. IKIM (Valiev & Tutubalina, 2024) employs in-prompt ensembling with entity-level signals and knowledge-graph constraints, generating multiple candidates and consolidating them to improve fidelity in both detection and correction. In addition, we compare against recent large language models, including Gemini 2.0 Flash (Google Research, 2025), Gemini 2.5 Flash Lite (Google Research, 2025), GPT-4o-mini (OpenAI, 2024), Doubao-1.5-thinking-pro (Doubao Team, Volcano Engine, 2024), and Deepseek-V3 (DeepSeek AI, 2024).

ROUGE-1 measures lexical overlap via unigram recall. BERTScore captures semantic similarity using contextualized embeddings. BLEURT, fine-tuned on human ratings, evaluates both fluency and semantic adequacy. Our proposed KPCS metric further addresses domain-specific evaluation needs by incorporating medical factual consistency and terminology alignment, better reflecting correction quality in the clinical context.

## A.2 Experimental settings

All experiments were run using the following large-language models: Gemini 2.0 Flash[1], gemini-2.5-flash-lite[2], GPT-4o mini[3], doubao1.5-thinking-pro[4], and deepseek-v3[5]. For gemini-2.0-flash, gemini-2.5-flash-lite, doubao1.5-thinking-pro and deepseek-v3 we used each model's provider default API parameters and recorded the raw scores produced by the model calls. For GPT-4o mini we fixed the decoding parameters to `temperature=0.2`, `top_p=1.0`, `max_tokens=512`, and `logprobs=True`. DeepSeek (the `deepseek-v3-250324` checkpoint) was invoked via the Doubao API interface in our setup.

To map a model prediction that identifies an erroneous sentence back to the original document, we used the Longest Common Subsequence (LCS) between the predicted text span and each candidate sentence in the original document and selected the sentence with the highest LCS score as the located error sentence. For metric computation we report Error Sentence Location Accuracy, ROUGE-1, BERTScore, BLEURT and KPCS. When computing these metrics, cases in which the original sentence had no error and the model also predicted no error (true negatives) were counted as successful cases and included in the final scores. All reported numeric results were rounded to four decimal places. We asked models to expose structured outputs to simplify automatic parsing. Specifically, we encouraged models to emit the chain of thought inside `<think>(.*?)</think>` tags, a confidence score inside `<confidence>(\d{1,3})</confidence>` tags, and the final decision inside `<result>(.*?)</result>` tags. We then extracted those fields by regular-expression matching and used the extracted values in downstream evaluation and analysis. In our framework, self-consistency during the detection and localization stages is achieved by employing multiple models with distinct prompts, rather than relying on repeated prompting of a single model. Each model separately generates its output, and their results are compared to reach a consistent decision through arbitration if disagreement occurs.

## A.3 Human evaluation

To evaluate the effectiveness of `MedGuards`, we conducted an evaluation involving two clinical doctors to calculate the Mean Opinion Score (MOS). Using the MEDEC dataset, we selected Gemini 2.0 Flash and GPT-4o-mini as the base models and assessed the impact of applying `MedGuards`. For each set of results, we randomly sampled 20 cases, resulting in a total of $20 \times 4 = 80$ cases for evaluation. Each case was rated on a 4-point scale (0–3), where 0 indicates a completely incorrect correction (e.g., incorrect diagnosis or insufficient diagnostic evidence), 3 indicates excellent correction performance, and 1 and 2 represent intermediate levels of correction quality. For comparison with other evaluation metrics, the final MOS values were normalized to the range [0, 1]. The rating shows a 57.5% agreement, reflecting the consistency among clinicians while still allowing for reasonable variability. The agreement was computed as the proportion of cases where both clinicians assigned the same score to the corresponding model outputs for a given case. Since each clinician rated cases based on their own medical experience and judgment, some variation across ratings is expected and aligns with real-world practice.

Table 6 reports human evaluation results on the MEDEC dataset. We observe that `MedGuards` consistently achieves higher scores than its corresponding base models, confirming that our method improves correction quality. This outcome resonates with the design of the KPCS metric, as `MedGuards` demonstrates a stronger ability to capture and preserve critical keyword information, which human evaluators also recognize as central to correction quality.

---

[1]`https://cloud.google.com/vertex-ai/generative-ai/docs/models/gemini/2-0-flash`
[2]`gemini-2.5-flash-lite-preview-06-17`, `https://cloud.google.com/vertex-ai/generative-ai/docs/models/gemini/2-5-flash-lite`
[3]`https://openai.com/index/gpt-4o-mini-advancing-cost-efficient-intelligence/`
[4]`doubao-1-5-thinking-pro-250415`, `https://www.ohmygpt.com/pricing/model/doubao-1.5-thinking-pro-250415`
[5]`deepseek-v3-250324`, `https://huggingface.co/deepseek-ai/DeepSeek-V3-0324`

Table 6: Human evaluation results on the MEDEC dataset. Scores range from 0 to 3, with higher scores indicating better correction quality. The final MOS values were normalized to the range [0, 1].

| Model | Normalized MOS Score |
|---|---|
| Gemini 2.0 Flash | 0.425 |
| + MedGuards | 0.483 |
| GPT-4o-mini | 0.383 |
| + MedGuards | 0.475 |

## A.4 ADDITIONAL RESULTS ON SELF-CONSISTENCY RATE

Figures 4, 5, and 6 show the proportions of Consensus versus Arbitration within the self-consistency framework across different base models and stages of the pipeline on the multilingual dataset. Across all three datasets, we observe that consensus dominates the outcomes, indicating that agents typically reach agreement without arbitration. In particular, for error detection tasks, on all three datasets and for both models, the arbitration rates range from 11.5% to 20.2%; while localization tasks have wider ranges, that is from 7.9% to 28.6%. On English, Gemini has a higher arbitration rate for detection, but similar rate for localization compared to GPT; while this phenomenon is totally opposite on Arabic. In general, both Gemini and GPT exhibit a comparable likelihood of arbitration, suggesting that the two models have higher or lower arbitration rates over each other in different cases and reflecting their distinct consistency across scenarios.

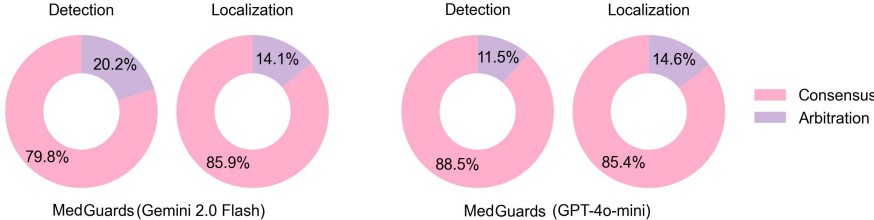

Figure 4: Distribution of self-consistency rate on the English dataset of our internal test set.

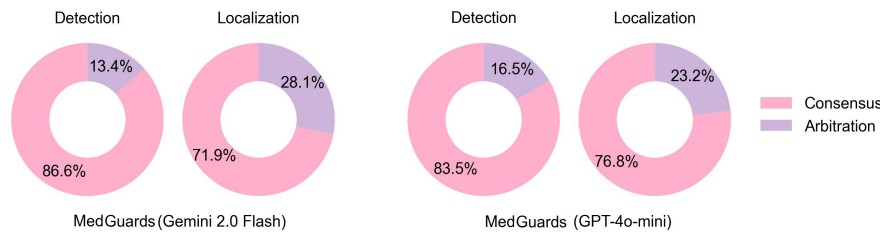

Figure 5: Distribution of self-consistency rate on the Arabic dataset of our internal test set.

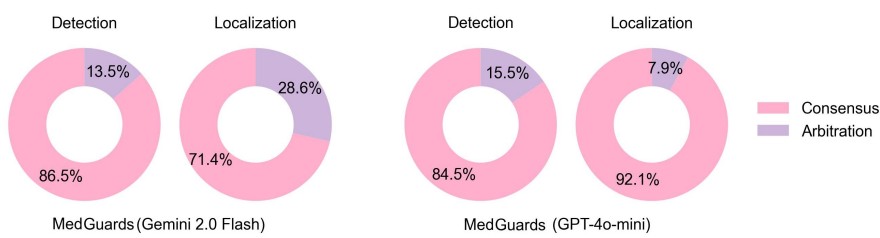

Figure 6: Distribution of self-consistency rate on the Chinese dataset of our internal test set

## A.5 MIXTURE OF AGENTS

We conducted experiments covering representative configurations across the three stages on the MEDEC dataset, including: (a) variations where different models are mixed within one stage, and (b) combinations where the different stages use distinct models. The results are shown in Tables 7, 8 and 9. Tables 7 and 8 use Gpt-4o-mini as the base model for `MedGuards`.

In Table 7, most other model mixtures lead to slightly lower performance. However, we also observe an interesting case: when Gemini 2.0 Flash is introduced as an additional agent in the localization stage (serving as an agreement agent), the localization and correction performance increase. This suggests that combining complementary LLMs can bring marginal gains in specific sub-tasks but requires further investigation.

Table 7: Mixed-model configurations within one stage on the MEDEC dataset using GPT-4o-mini and Gemini 2.0 Flash, denoted as GPT and Ge in the table without loss of brevity. We assume that the correction agent is always GPT-4o-mini. (2 GPT + 1 Ge): the two initial agents are GPT-4o-mini, and the arbitration agent is Gemini 2.0 Flash. (1 GPT + 1 Ge) + 1 Ge: the two initial agents are GPT-4o-mini and Gemini 2.0 Flash, with Gemini 2.0 Flash as the arbitration agent.

| Detection Agent | Localization Agent | Detection Accuracy | Localization Accuracy | Correction ROUGE-1 | Correction BERTScore | Correction BLEURT |
|---|---|---|---|---|---|---|
| 3 GPT | 3 GPT | **0.695** | 0.535 | 0.571 | **0.581** | **0.574** |
| 2 GPT + 1 Ge | 3 GPT | 0.652 | 0.563 | 0.506 | 0.509 | 0.504 |
| (1 GPT + 1 Ge) + 1 Ge | 3 GPT | 0.674 | 0.576 | 0.517 | 0.521 | 0.515 |
| 3 GPT | 2 GPT + 1 Ge | **0.695** | 0.580 | 0.518 | 0.521 | 0.512 |
| 3 GPT | (1 GPT + 1 Ge) + 1 Ge | **0.695** | **0.638** | **0.577** | 0.577 | 0.565 |

Tables 8 and 9 compare using distinct models (single type) across the different stages. We observe that using stronger detection agents as base models, such as Doubao-1.5-thinking-pro, can improve model performance. For instance, pairing Doubao with Gemini 2.0 Flash achieves the highest scores across all metrics. Pairing Doubao with GPT-4o-mini produces much better performance in error detection and localization, but not for the other metrics.

Table 8: Performance comparison of different agent configurations on the MEDEC dataset.

| Detection Agent | Localization Agent | Correction Agent | Detection Accuracy | Localization Accuracy | Correction ROUGE-1 | Correction BERTScore | Correction BLEURT |
|---|---|---|---|---|---|---|---|
| GPT-4o-mini | GPT-4o-mini | GPT-4o-mini | 0.695 | 0.535 | **0.571** | **0.581** | **0.574** |
| GPT-4o-mini | Gemini 2.0 Flash | GPT-4o-mini | 0.695 | 0.583 | 0.543 | 0.543 | 0.533 |
| GPT-4o-mini | GPT-4o-mini | Gemini 2.0 Flash | 0.695 | 0.535 | 0.563 | 0.580 | 0.502 |
| Gemini 2.0 Flash | GPT-4o-mini | GPT-4o-mini | **0.713** | 0.484 | 0.547 | 0.536 | 0.573 |
| Gemini 2.0 Flash | Gemini 2.0 Flash | Gemini 2.0 Flash | **0.713** | **0.609** | 0.537 | 0.549 | 0.549 |

Table 9: Performance comparison of using different detection and correction agent configurations on the MEDEC dataset. The localization agent is assumed to be Gemini 2.0 Flash.

| Detection Agent | Correction Agent | Detection Accuracy | Localization Accuracy | Correction ROUGE-1 | Correction BERTScore | Correction BLEURT |
|---|---|---|---|---|---|---|
| Gemini 2.0 Flash | Gemini 2.0 Flash | 0.713 | 0.609 | 0.537 | 0.549 | 0.549 |
| Doubao-1.5-thinking-pro | Gemini 2.0 Flash | **0.770** | **0.669** | **0.591** | **0.591** | **0.600** |
| GPT-4o-mini | GPT-4o-mini | 0.695 | 0.535 | **0.571** | **0.581** | **0.574** |
| Doubao-1.5-thinking-pro | GPT-4o-mini | **0.770** | **0.611** | 0.559 | 0.570 | 0.500 |

## A.6 LATENCY AND COMPUTATIONAL COST ANALYSIS

To quantitatively assess the computational overhead introduced by MedGuards, we measured the average inference time per clinical note, as well as the average number of input and output tokens across all agents on the MEDEC dataset. We adopted Gemini 2.0 Flash, where the rate is approximately $0.10 per 1M input tokens, $0.40 per 1M output tokens, and $0.025 per 1M tokens with context caching. The results are shown in Table 10. While our proposed multi-agent framework `MedGuards` introduces an increase in latency due to multiple specialized reasoning steps and API

calls, the accuracy improvement is crucial in time-sensitive clinical scenarios where decision reliability outweighs minor latency increases. In practical deployment, techniques such as agent parallelization and context caching can further mitigate latency without compromising performance.

Table 10: Latency and Computational Cost Analysis.

| Models | Latency (s) | Input tokens (N) | Output tokens (N) | Performance improvement (%) |
|---|---|---|---|---|
| Gemini 2.0 Flash | 1.61 | 1927 | 48 | – |
| + MedGuards | 6.99 | Detection: 1047 Localization: 1127 Correction: 1135 | Detection: 1159 Localization: 666 Correction: 623 | **96.10%** |

### A.7 ANALYSIS OF THE MULTI-AGENT FRAMEWORK STRUCTURE

To further investigate the impact of structural design on multi-agent performance, we conduct an ablation study that examines different configurations of the multi-agent framework. We use Gemini 2.0 Flash as the base model. We vary the number and role allocation of agents and report the results in Table 11. The number of agents ranges from 1 to 8. In the two-agent correction setting, the first agent generates an initial correction based on the model's output. This intermediate result is then passed to the second agent, which produces the final answer by considering both the original input and the first agent's response. Overall, the detection and localization stages tend to show improved performance, although this trend is not strictly monotonic. For example, using one agent for each of the three stages performs worse than sharing a single agent between detection and localization. Compared to the single-agent baseline, the two-agent setting yields a 17.6% improvement in the overall average score, and expanding to four agents further increases the gain to 28.9%. When both detection and localization adopt three agents with arbitration, the performance reaches its best improvement of 36.8%. In contrast, the correction stage exhibits an opposite trend: increasing the number of correction agents leads to a drop in generation quality, with performance falling to only a 12.5% improvement when two correction agents are used. This is likely because correction is a generation task, where multiple agents may produce different valid outputs that are difficult to unify, making consensus-based arbitration challenging. The results show that our default MedGuards configuration, three agents for detection, three for localization, and one for correction—achieves the best overall performance across all metrics.

Table 11: Performance comparison across different agent configurations. $\Delta\%$ shows average improvement across all metrics over the one agent for all task.

| Detection Agents (N) | Localization Agents (N) | Correction Agents (N) | Total Agents (N) | Detection Accuracy | Localization Accuracy | Correction ROUGE-1 | Correction BERTScore | Correction BLEURT | Average Score | $\Delta\%$ |
|---|---|---|---|---|---|---|---|---|---|---|
| 1 Agent for all tasks | | | 1 | 0.589 | 0.415 | 0.379 | 0.380 | 0.395 | 0.432 | – |
| 1 Agent for detection & localization | | 1 | 2 | 0.688 | 0.484 | 0.410 | 0.446 | 0.511 | 0.508 | 17.6% |
| 1 | 1 | 1 | 3 | 0.650 | 0.250 | 0.439 | 0.416 | 0.513 | 0.454 | 5.09% |
| 3 | 1 Agent for localization & correction | | 4 | **0.713** | 0.501 | 0.535 | 0.521 | 0.515 | 0.557 | 28.9% |
| 3 | 3 | 1 | 7 | **0.713** | **0.609** | **0.537** | **0.549** | **0.549** | **0.591** | 36.8% |
| 3 | 3 | 2 | 8 | **0.713** | **0.609** | 0.367 | 0.367 | 0.374 | 0.486 | 12.5% |

### A.8 SENSITIVITY ANALYSIS OF THE BALANCE FACTOR $\alpha$

The balance factor $\alpha$ is set to 0.5 in all experiments. To evaluate the sensitivity of the balance factor $\alpha$, we varied the weighting parameter $\alpha$ in the KPCS metric to 0.2, 0.5, and 0.8, and report the results in Table 12. As shown in the table, varying $\alpha$ does not lead to substantial changes in overall model ranking, but the margins between models shift because KPCS rebalances important-word vs. overall semantics as $\alpha$ changes A smaller $\alpha$ emphasizes overall semantics, while a larger $\alpha$ gives more weight to keywords. The results indicate that $\alpha = 0.5$ yields KPCS scores most consistent with human evaluation (in Table 6) trends. For both Gemini 2.0 Flash and GPT-4o-mini, the model ranking under $\alpha = 0.5$ best matches the human Normalized MOS results, while $\alpha = 0.2$ or 0.8 tends to over or under emphasize differences.

Table 12: Performance comparison under different Alpha values in KPCS.

| Alpha | Gemini 2.0 Flash + MedGuards | GPT-4o-mini + MedGuards | Doubao-1.5-thinking-pro + MedGuards | Deepseek-v3 + MedGuards | Gemini 2.5 Flash Lite + MedGuards |
|-------|------------------------------|-------------------------|-------------------------------------|-------------------------|-----------------------------------|
| 0.2   | 0.516                        | 0.524                   | 0.594                               | 0.599                   | 0.370                             |
| 0.5   | 0.472                        | 0.461                   | 0.554                               | 0.556                   | 0.385                             |
| 0.8   | 0.428                        | 0.380                   | 0.513                               | 0.513                   | 0.294                             |

## A.9  QUALITATIVE BEHAVIOR OF THE AGENTS IN THE DETECTION & LOCALIZATION STAGES

We further analyze the qualitative behavior of the agents at each stage on MEDEC dataset. The results in Table 13 indicate that the lower-performing GPT-4o-mini + `MedGuards` model exhibits a higher rate of agent disagreement in both the detection and localization stages, leading to more frequent entries into the arbitration stage. Second, for both models, cases entering arbitration exhibit lower accuracy. The arbitration process still corrects about half of such cases, indicating partial complementarity between agents. Moreover, localization tasks generally show higher disagreement rates than detection, suggesting that fine-grained spatial reasoning remains more challenging. These analyses provide qualitative insights into the failure patterns of each LLM and demonstrate the robustness and diagnostic value of the `MedGuards` design.

Table 13: Behavior of the agents in the detection and localization stages.

| Models | Statistics | Detection | | Localization | |
|--------|-----------|-----------|-----------|--------------|-----------|
| | | Agreement | Arbitration | Agreement | Arbitration |
| Doubao-1.5-Thinking-Pro + MedGuards | Number of Cases (%) | 818 (88.43%) | 107 (11.57%) | 401 (86.61%) | 61 (13.20%) |
| | Correctness Rate (No. of Correct Cases) | 80.44% (658) | 46.73% (50) | 71.07% (285) | 49.18% (30) |
| | Error Rate (No. of Wrong Cases) | 19.56% (160) | 53.27% (57) | 28.93% (116) | 50.82% (31) |
| GPT-4o-mini + MedGuards | Number of Cases (%) | 725 (78.4%) | 200 (21.6%) | 274 (48.4%) | 292 (51.6%) |
| | Correctness Rate (No. of Correct Cases) | 69.24% (502) | 48.5% (97) | 47.4% (130) | 33.2% (97) |
| | Error Rate (No. of Wrong Cases)) | 30.76% (223) | 51.5% (103) | 52.6% (144) | 66.8% (195) |

## A.10  EXAMPLES OF THE ANNOTATED KEYWORDS

Table 14 shows examples of the annotated keywords from the MEDEC dataset. The domain-specific keywords are directly tied to error types, not to general clinical facts. For example, in MEDEC, the keywords correspond to five error categories: Diagnosis, Management, Treatment, Pharmacotherapy, and Causal Organism. Importantly, factual information such as patient symptoms or pre-existing conditions at admission is not included in the keywords, since we are concerned with the keywords that constitute an error. For instance, in the first clinical note in Table 14, "sepsis" is not treated as a keyword because it represents an existing diagnosis at admission ("*Mr. ¡NAME/¿ was treated for sepsis, present on admission*"), rather than an erroneous or modifiable statement.

## A.11  THE PROMPTS OF MEDGUARDS

The prompts used by MedGuard are shown in Table 15.

## A.12  ADDITIONAL DISCUSSION

In our current implementation, the first component of the KPCS score relies on exact string matching. While this approach ensures precision, it may overlook clinically equivalent expressions. Developing a more flexible alternative would require a principled strategy for handling synonyms, ideally through the integration of medical ontologies or structured clinical knowledge bases.

Table 14: Examples of the annotated keywords.

| Erroneous Clinical Note | Corrected Clinical Note | Keywords |
| --- | --- | --- |
| Mr. <NAME/> was Mr. <NAME/> treated for sepsis, present on admission. Pt is a <AGE/> yo M who was admitted with hypoxia, cough and AMS; pt is discovered to have pneumonia and blood cultures from admission grew Strep pneumococcus. Pt presented with WBC 29, tachypneic, with AKI and somnolence/lethargy. Mental status much improved and Fever and WBC coming down on hospital day #2 following IV abx and IVF; an 8 day course of **antifungal medication** at home will be completed. | Mr. <NAME/> was Mr. <NAME/> treated for sepsis, present on admission. Pt is a <AGE/> yo M who was admitted with hypoxia, cough and AMS; pt is discovered to have pneumonia and blood cultures from admission grew Strep pneumococcus. Pt presented with WBC 29, tachypneic, with AKI and somnolence/lethargy. Mental status much improved and Fever and WBC coming down on hospital day #2 following IV abx and IVF; an 8 day course of **antibiotics** at home will be completed. | **antibiotics** |
| Ms. <NAME/> is being managed for Stage 3 R and L coccyx pressure ulcers, present on admit (POA). Patient is a <AGE/> YO F recently discharged after a 1 month hospitalization for recurrent cirrhosis of transplant liver graft s/p TIPS, hepatic hydrothorax with VRE positive pleural fluid s/p removal of chest tubes, and pressure ulcers who presents with 24 hours of fever, rigors, nausea/vomiting. <DATE/> Wound RN - Patient has pressure ulcers over right and left coccyx. The right coccyx pressure ulcer is an unstageable, at least stage three, pressure ulcer. Left coccyx wound is unstageable, at least stage 3 pressure ulcer. <DATE/> RN IVIEW - IVIEW <DATE/> - R & L buttocks pressure ulcers Stage 3. <DATE/> Skin: several sacral decubitus ulcers: noninfected appearing. institute wound care per previous recommendations. <DATE/> SKIN: Buttock ulcers stage 3. Other treatments include: Patient on pressure reducing surface, **zinc citrate** moisture barrier, Assist patient to obtain adequate nutrition. | Ms. <NAME/> is being managed for Stage 3 R and L coccyx pressure ulcers, present on admit (POA). Patient is a <AGE/> YO F recently discharged after a 1 month hospitalization for recurrent cirrhosis of transplant liver graft s/p TIPS, hepatic hydrothorax with VRE positive pleural fluid s/p removal of chest tubes, and pressure ulcers who presents with 24 hours of fever, rigors, nausea/vomiting. <DATE/> Wound RN - Patient has pressure ulcers over right and left coccyx. The right coccyx pressure ulcer is an unstageable, at least stage three, pressure ulcer. Left coccyx wound is unstageable, at least stage 3 pressure ulcer. <DATE/> RN IVIEW - IVIEW <DATE/> - R & L buttocks pressure ulcers Stage 3. <DATE/> Skin: several sacral decubitus ulcers: noninfected appearing. institute wound care per previous recommendations. <DATE/> SKIN: Buttock ulcers stage 3. Other treatments include: Patient on pressure reducing surface, **zinc oxide** moisture barrier, Assist patient to obtain adequate nutrition. | **zinc oxide** |
| Mr. <NAME/> is a <AGE/>-old man with gastrinoma metastatic to liver s/p Exploratory laparotomy, intra-operative ultrasound, partial hepatectomy (segment 4B, 5, 7), RFA of liver tumors (segment 3, 8), distal pancreatectomy and splenectomy on <DATE/>. Surgery note <DATE/> : New O2 requirement, CXR read as atelectasis or pneumoni with low lung volumes. <DATE/> Surgery note - PE w/ increased O2 requirement, tachypnea, currently on high flow 100% at 50L. <DATE/> Icu note: Respiratory distress, secondary to PE, concern for respiratory failure. Treatment CXR, admit to ICU, Continue **Lepirudin** drip, Continue HFNC, Repeat ABG to determine worsening/stability of hypoxemia. This patient also being managed for Acute hypoxemic respiratory insufficiency - multifactorial, from volume overload, hypoventilation from ileus, and from PE. | Mr. <NAME/> is a <AGE/>-old man with gastrinoma metastatic to liver s/p Exploratory laparotomy, intra-operative ultrasound, partial hepatectomy (segment 4B, 5, 7), RFA of liver tumors (segment 3, 8), distal pancreatectomy and splenectomy on <DATE/>. Surgery note <DATE/> : New O2 requirement, CXR read as atelectasis or pneumoni with low lung volumes. <DATE/> Surgery note - PE w/ increased O2 requirement, tachypnea, currently on high flow 100% at 50L. <DATE/> Icu note: Respiratory distress, secondary to PE, concern for respiratory failure. Treatment CXR, admit to ICU, Continue **heparin** drip, Continue HFNC, Repeat ABG to determine worsening/stability of hypoxemia. This patient also being managed for Acute hypoxemic respiratory insufficiency - multifactorial, from volume overload, hypoventilation from ileus, and from PE. | **heparin** |

Given that structured generation with XML-like tags has proven effective and stable in our setup, an interesting future direction is to explore more principled or model-agnostic ways to ensure structural reliability, such as schema-constrained decoding or adaptive validation mechanisms.

Our current work is methodology-oriented and serves primarily as a proof of concept, focusing on the feasibility of the proposed approach rather than immediate clinical integration. In future work,

Table 15: Prompts for `MedGuards` on MEDEC dataset. SC indicates self-consistency.

| Prompt Name | Prompt Content |
|---|---|
| **Detection prompt A** | The following is a medical narrative about a patient. You are a skilled medical doctor reviewing the clinical text. The text is either correct or has at most a medical error related to treatment, management, cause, diagnosis or causalOrganism. Write down your thinking process in <think> thinking process here </think> tags. Check every sentence of the text. If the text is correct return one word "CORRECT". If the text has a medical error return one word "INCORRECT". Also output your confidence in <confidence>(1-100 score)</confidence> tags. |
| **Detection prompt B** | You are a skilled medical doctor reviewing the clinical text. The text is either correct or has a medical error related to diagnosis (put more focus), treatment, management, cause or causalOrganism. Write down your thinking process in <think> thinking process here </think> tags. Check every sentence of the text. If the text is correct return one word "CORRECT". If the text has a medical error return one word "INCORRECT". Also output your confidence in <confidence>(1-100 score)</confidence> tags. |
| **Detection SC prompt** | You are a medical expert reviewing the clinical text and two junior doctors. The text is either accurate or contains at most a medical error, primarily related to diagnosis (pay closer attention), but may also involve treatment, management, cause, or causal organism. Document your reasoning within <think> your reasoning here </think> tags. Evaluate each sentence in the text. If the text is accurate, return the single word "CORRECT". If it contains a medical error, return the single word "INCORRECT". Also output your confidence in <confidence>(1-100 score)</confidence> tags. |
| **Localization prompt A** | You are {agent_name}, a medical report quality assurance expert. The report is either correct or has at most a medical error related to treatment, management, cause, diagnosis or causalOrganism. Your task is to identify the **one** sentence that contains an error (or there are zero errors return 'NAN') in the following medical report and put your prediction in <result>Ensure you have results here</result> tags. Document your reasoning within <think></think> tags. A useful tip is that the error sentence (if there is) is more likely to be found in a conclusion sentence. Please ONLY return the original erroneous sentence exactly as it appears in the text (the text may also contain zero errors. If you are sure about it, please return 'NAN'). Also output your confidence in <confidence>(1-100 score)</confidence> tags. |
| **Localization prompt B** | You are {agent_name}, a medical report quality assurance expert. The report is either correct or has at most a medical error related to treatment, management, cause, diagnosis or causalOrganism. Your task is to identify the one sentence that contains an error (or there are zero errors return 'NAN') in the following medical report. A useful tip is that the error sentence (if there is) is more likely to be found in a conclusion sentence. Put your prediction in <result>Ensure you have results here</result> tags and document your reasoning within <think></think> tags. Also output your confidence in <confidence>(1-100 score)</confidence> tags. If the text has an error, output: <result>The Error Sentence</result>. If the text has no error, output: NAN. |
| **Localization SC prompt** | You are a senior medical report quality assurance expert. The report is either correct or has at most a medical error related to treatment, management, cause, diagnosis or causalOrganism. Here is the full report: {full_text}. Your 2 colleagues suggest that the following sentence is erroneous with reasons: "{partner_opinion}". Based on the above information, please provide the ONLY error sentence **you** believe (or there are zero errors return 'NAN'). Please put results in <result>Ensure you have results here</result> tags and document your reasoning within <think></think> tags. A useful tip is that the error sentence (if there is) is more likely to be found in a conclusion sentence. Also output your confidence in <confidence>(1-100 score)</confidence> tags. |
| **Correction prompt** | You are a medical report quality assurance expert. The following sentence is incorrect and has a medical error related to treatment, management, cause, diagnosis or causalOrganism. Please provide a corrected version to <result>the corrected sentence</result> tags (your thinking reason can be provided in <think></think> tags). Also, ensure you output results between <result></result> tags and output your confidence in <confidence>(1-100 score)</confidence> tags. Here is the original full report: {full_text}. Predicted Erroneous Sentence: {error_sentence}. Corrected Sentence: |

we plan to collaborate with clinical partners to evaluate the framework in controlled studies, systematically assessing its safety, usability, and compliance. Importantly, the framework is designed to support human-in-the-loop workflows, ensuring that clinicians can review and validate system outputs before use. Beyond documentation generation, such a setup could also facilitate downstream tasks such as error detection, quality audits, and clinical data validation.

In relation to the choice between open-source and proprietary models, we view this consideration as orthogonal to the core contribution of our work. The proposed framework is inherently model-agnostic, meaning it can be seamlessly applied to both open-source and proprietary LLMs, thereby providing flexibility across different deployment scenarios.

While our study demonstrates the feasibility of automated medical error detection and correction using a controlled and well-defined benchmark dataset, an important direction for future work is the incorporation of real-world clinical data. At present, annotated datasets for this task are scarce, as the research area is still emerging and the creation of high-quality annotations requires substantial clinical expertise. Moving forward, we plan to collaborate closely with clinical practitioners to collect real-world clinical data, which we believe will enable a more comprehensive evaluation of our method and help bridge the gap between research and clinical application.

### A.13 CONFIDENCE ANALYSIS IN THE DETECTION STAGE

On the MEDEC dataset, we performed a confidence score analysis using the best performing `MedGuards` model (based on Doubao-1.5-thinking-pro) focusing on agreement cases. We compared confidence scores between correct and wrong predictions. As shown in Table 16, we observe that the confidence scores are not directly related to whether a case is correct or incorrect. Among the agreement cases in the detection stage, the numbers of instances where both agents' confidence scores are simultaneously: $<90$ is 18 (1.95%); $<85$ is 2 (0.22%) and $<80$ is 0 (0%). In cases where both agents have confidence scores below 85, the two agents agree 100% of the time, and all predictions are correct. Therefore, in this setting, the arbitration agent does not need to intervene for the low-confidence samples. In cases where both agents have confidence scores below 90, the agents agree in (17/18) 94.4% of instances, and all of these predictions are correct. In the remaining (1/18) 5.6% of instances, the two agents disagree. After entering the arbitration, the system makes an incorrect prediction. Hence, under this setting, the arbitration agent did not need to intervene for two low-confidence samples either.

Table 16: Average confidence score analysis on MEDEC dataset.

| Average confidence score | Detection - agent 1 | Detection - agent 2 |
|---|---|---|
| All cases | 96.3 | 95.8 |
| Correct cases | 96.2 | 95.8 |
| Wrong cases | 96.4 | 95.5 |

