# OpenReview forum: "MedGuards: Multi-Agent System for Reliable Medical Error Detection and Correction"
_ICLR.cc/2026/Conference — Submitted to ICLR 2026_

### Official Review · Reviewer_wD7m · 2025-10-26

**Soundness:** 2
**Presentation:** 3
**Contribution:** 3
**Rating:** 6
**Confidence:** 3

**Summary:**

The paper proposes "MedGuards" -- a framework which employs a multi-agentic system for medical report generation error detection and correction. The paper additionally proposes a new metric KPCS which aims to overcome the shortcomings of alternative popular metrics by assigning a higher weight to more important words in the generated text. The methodology demonstrates substaintial gains in performance over baseline methods with propriatary LLMs.

**Strengths:**

* The work is timely and important. A good solution could have an important impact on real-world environment as healthcare. Agentic AI is a growing and popular area of research.
* The proposed methodology provides good performance increases with a relatively simple implementation.
* The paper is generally written to a good standard, but some key information and citations are missing (see weaknesses).
* Thorough ablation studies

**Weaknesses:**

* **The writing is lacking sufficient evidence for claims. The writing can be notably improved** For example (this is just the first page):
    * "Existing applications of LLMs in medical text generation often lack robust error correction mechanisms, a gap that is particularly critical in high-stakes clinical settings." (cf. line 39) -- citation required
    * "Current quality control methods predominantly focus on assessing outputs on the semantic level, which is insufficient for addressing errors that require deep structural understanding and clinical reasoning, both essential in medical scenarios (Li et al., 2024)." (cf. line 41) -- the citation is a benchmark for evaluation Q&A ability of LLMs in medicine. This is insufficient for the claim the current QC methods predominantely focus on outputs on semantic level, for example a quick google scholar shows methods such as deferral [1]. Additionally, you must provide evidence evidence for the claim "essential in medical scenarios"
   * "medical text often involves subtle errors" (cf. line 46) -- citation required, "...which is naturally aligned with the scenarios paradigm and the idea of Chain-of-Though" -- why? (+citation required)
  * "complex tasks are more tractable when decomposed into interpretable intermediate steps" (cf. line 49) -- citation required
  * "medical applications demand high reliability, which a single model output cannot guarantee" (cf. line 50) -- citation required. How are you defining reliablity? This is a very bold claim (that a single model output cannot guarantee). There are many examples of a single model being deployed in healthcare (of which we can assume after passing strict regulations & laws, are sufficiently reliable)
* **It would be nice for a central figure to help readers understand the pipleline of the proposed methodology**
* **Some information may be missing**:
     * What are the confidence scores? How are they derived? Are they calibrated?
     * How are the domain-specific keywords for KPCS determined/derived? Is it just a fixed list of words? What happens if some text contains a critical word not on this list?
* **No LLMs applicable for the use in healthcare are benchmarked**. I think it's important to place the analysis in the context of the authors' proposed setting of healthcare. Here, there are stringent regulations regarding data privacy to protect patients and health care providers. In my own experience, it would be impossible to use such proprietary LLMs used in the analysis of this paper for these reasons. A better evaluation would additionally benchmark against open-source medical LLMs.
* **Consideration of application in healthcare**. I think the paper sorely misses discussing the application of the proposed method in healthcare, of which comes with strigent regulations. The authors motivate the setting of healthcare as it's important to mitigate errors here (for patient safety), but then do not discuss the implications of the proposed method on patient safety.
* **Technical novelty** - I am somewhat concerned that the technical novelty may not pass the high bar required for ICLR

Others:
* The paper claims the proposed method increases ``reliability'' (of which I'm assuming the authors are defining as the reduction of errors in generated medical text -- though no definition is given). In doing so, I really think the authors should be reporting variability in performance over the multiple runs.

Minor:
* "multiple specialized agents that compute predictions independently" (many places in the paper) I think a better word is "separately". Independently has mathematical connotations of which I do not think apply here.


Citations:
[1] Strong et al. 2019 "Trustworthy and Practical AI for Healthcare: A Guided Deferral System with Large Language Models" (AAAI-25)

**Questions:**

1. What are the confidence scores? How are they derived? Are they calibrated?
2. How are the domain-specific keywords for KPCS determined/derived? Is it just a fixed list of words? What happens if some text contains a critical word not on this list?
3. Can you comment on the applicability of MedGuards for the real-world deployment in healthcare? What are the computational/monetary costs? Is the error detection/localization/correction performance sufficient enough for use? Do you not need to benchmark against open-source LLMs?

---

> ### Author Response · Authors · 2025-11-23
> **Response to Reviewer wD7m --- Part 1**
>
> Thank you for your constructive feedback! Below is our detailed response.
>
> > The writing is lacking sufficient evidence for claims
>
> Thank you for your suggestions. We added the citations and explanations in the updated version of the manuscript.
>
> > It would be nice for a central figure to help readers understand the pipeline of the proposed methodology
>
> Thank you for your suggestion, we created a high-level overview Figure that’s now incorporated in the updated manuscript.
>
> > What are the confidence scores? How are they derived? Are they calibrated?
>
> The confidence scores are generated by the LLMs. Each model is prompted to output its confidence within <confidence>(1–100)</confidence> tags by using the prompt “Output your confidence in <confidence>(1–100 score)</confidence> tags.”). These scores reflect the model’s own estimated certainty and are not additionally calibrated.
>
>
>
> > How are the domain-specific keywords for KPCS determined/derived? Is it just a fixed list of words? What happens if some text contains a critical word not on this list?
>
> The domain-specific keywords refer to key clinical concepts, such as diagnoses, medications, therapeutic actions, or essential findings, that are central to the clinical meaning of each note and particularly relevant to potential errors. These keywords are not from a fixed predefined list but rather specific to each sample. They were manually annotated by NLP researchers and verified by clinical experts across data samples, ensuring comprehensive coverage of important clinical terms. Consequently, every note in the datasets has been annotated, and thus the issue of missing critical keywords does not occur in our current setting.
>
> > Consideration of application in healthcare. I think the paper sorely misses discussing the application of the proposed method in healthcare, of which comes with strigent regulations. The authors motivate the setting of healthcare as it's important to mitigate errors here (for patient safety), but then do not discuss the implications of the proposed method on patient safety.
>
> We appreciate this point. Our current work is methodology-oriented and acts as a proof-of-concept focusing on methodological feasibility rather than direct clinical deployment. We fully agree that real-world integration in healthcare requires careful consideration of regulatory compliance (e.g., HIPAA/GDPR) and patient safety. The proposed multi-agent framework is designed to support human-in-the-loop workflows, ensuring that clinicians can review and validate all system outputs before use. In the revised manuscript, we added a discussion paragraph highlighting the future implications of our framework for error correction and detection in clinical documentation (e.g., supporting audits), and we plan to conduct clinical studies in the future with clinical partners to systematically evaluate safety, usability, and compliance in a controlled setting.
>
> > No LLMs applicable for the use in healthcare are benchmarked.
>
> We evaluated MedGemma, a state-of-the-art open-source medical LLM for this purpose on the MEDEC benchmark:
>
> | Model                 | Detection Accuracy | Localization Accuracy | Correction ROUGE-1 | Correction BERTScore | Correction BLEURT |
> |------------------------|-------------------:|-----------------------:|-------------------:|---------------------:|------------------:|
> | MedGemma               | 0.507              | 0.478                  | 0.501              | 0.502                | 0.504             |
> | MedGemma + MedGuards   | 0.520              | 0.492                  | 0.503              | 0.503                | 0.544             |
>
> However, MedGemma achieves around 0.5 across metrics mainly because it predicts most clinical notes as “error-free.” This leads to a seemingly moderate accuracy increase but shows that the model fails to actually detect or correct real errors. The main challenge with medical LLMs is that adapting them to narrow clinical tasks often results in excessive task specialization, limiting their ability to generalize across broader medical reasoning skills [1].
>
> Regarding the use of open-source versus proprietary models, we view this as a broader point that is orthogonal to the significance of our proposed methodological framework. Our framework is model-agnostic and can be applied on top of either open-source or proprietary LLMs, offering flexibility in deployment settings. For our baseline models, we selected the highest-performing models based on initial empirical analysis to ensure a fair and representative evaluation. We added this to the discussion section.
>
> [1] Dorfner, Felix J., Amin Dada, Felix Busch, Marcus R. Makowski, Tianyu Han, Daniel Truhn, Jens Kleesiek et al. "Biomedical large languages models seem not to be superior to generalist models on unseen medical data." arXiv preprint arXiv:2408.13833 (2024).

---

> ### Author Response · Authors · 2025-11-23
> **Response to Reviewer wD7m --- Part 2**
>
> > Technical novelty
>
> We would like to emphasize that our work introduces substantive novelty in system design, decision coordination and model evaluation.
>
> (1) We present MedGuards, the first multi-agent system tailored for medical error detection and correction. While prior studies addressed error detection or correction separately, our system integrates detection, localization, and correction within a collaborative agent framework, enabling explicit division of roles, interpretable reasoning, and plug-and-play integration with existing LLM agents.
>
> (2) We propose a confidence-guided adjudication mechanism, where a third agent adjudicates disagreements based on reasoning traces and confidence scores, improving robustness and reliability under uncertainty.
>
> (3) We further introduce the first evaluation metric KPCS designed specifically for this task, which prioritizes domain-critical medical keywords, aligning evaluation with clinical safety requirements.
>
> We would also like to note that the ICLR review process encourages a broad definition of originality and significance, such as new problem formulations, combinations of existing ideas, novel applications, and enhancing existing results, which we believe our work directly provides.
>
> > The paper claims the proposed method increases ``reliability''. In doing so, I really think the authors should be reporting variability in performance over the multiple runs.
>
> We thank the reviewer for this valuable suggestion. In our paper, reliability refers to the consistency and correctness of generated clinical notes, operationalized as reduced error rates in detection, localization, and correction tasks. To explicitly address this concern, we conducted additional experiments over 5 independent runs and reported both the mean ± standard deviation and statistical significance (two-tailed t-test). The results on the MEDEC benchmark are summarized below (example for GPT-4o-mini):
>
> | Model        | Detection Accuracy | Localization Accuracy | Correction ROUGE-1 | Correction BERTScore | Correction BLEURT |
> |---------------|-------------------:|-----------------------:|-------------------:|---------------------:|------------------:|
> | GPT-4o-mini   | 0.661 ± 0.020     | 0.551 ± 0.011         | 0.589 ± 0.015     | 0.598 ± 0.013       | 0.582 ± 0.009    |
>
> We also report p-values to assess whether the improvements brought by MedGuards are statistically significant:
>
> | Model                     | p-value    |
> |----------------------------|------------:|
> | Gemini 2.0 Flash           | 4.03E-04   |
> | GPT-4o-mini                | 8.31E-05   |
> | Doubao-1.5-thinking-pro    | 1.33E-02   |
> | Deepseek-V3                | 3.15E-04   |
> | Gemini 2.5 Flash Lite      | 5.69E-03   |
>
> All results indicate statistically significant improvements (p < 0.05), demonstrating that the proposed MedGuards framework reliably enhances model performance.
>
> > "Multiple specialized agents that compute predictions independently" (many places in the paper) I think a better word is "separately". Independently has mathematical connotations of which I do not think apply here.
>
> Thank you for your suggestion. We replaced “independently” with “separately” for clarity.
>
> We kindly ask you to evaluate our submission in light of the above feedback and improvements made to our manuscript. Thank you for your time and effort.

---

> ### Comment · Reviewer_wD7m · 2025-11-25
>
> thanks for your rebuttal. I still remain a little unconvinced (apologies for the formatting):
>
> * The writing still feels to me as if it falls slightly below the bar. In some places, I have to use more effort than I would like to try and understand the message that the authors are trying to convey. I appreciate that the authors have added citations to add evidence to their claims, but having checked just a few, they don't seem to fully support the claims. For example, "First, medical text often involves subtle errors (Weiner & Schwartz, 2023; Behore et al., 2024)" -- this is an intuitive claim, but I don't think either citations provides evidence of this claim, especially in the medical context of the paper. Additionally, there still exists typos and formatting issues. For example, on line 277 the paper states that examples of the keywords are in Appendix A.9 (incorrect); line 285 "donote" and n and m should be $n$ and $m$. I haven't rigorously checked the entire paper, and finding these issues does not help me feel confident that there are not any others that I have not spotted.
>
> * The central Figure is good.
>
> * The domain-specific keywords still gives me some issues. From my perspective, the authors are too vague on what these keywords actually are. It seems to be a central part of the paper, so I'd expect this to be detailed explicitly. They state that the keywords are case-specific, but does that mean if one were to implement this in practice, you would need experts to annotate each test case for these keywords anyway? Therefore, during this process, wouldn't they spot the issues themselves? What's then the point of the methodology? I am confused on this aspect. Additionally, the authors provide examples in Table 14 - but surely isn't "sepsis" (and many others) a keyword for the first case?
>
> * Regarding the use of open-source versus proprietary models, "we view this as a broader point that is orthogonal to the significance of our proposed methodological framework". I have to disagree on this point. From my anecdotal experience, in the medical domain, the type of models (i.e., open-source versus proprietary) is highly relevant and therefore not independent of the proposed methodological framework, which deeply roots itself within the healthcare domain. It is unlikely to allow sensitive patient data be sent to 3rd parties to access proprietary. It is my opinion that a more thorough benchmarking against open-source models (doesn't have to be medical specialized) is missed.
>
> * I am still a little concerned that the paper passes the extremely high technical bar set by ICLR standards. The authors state ".. our work introduces substantive novelty in system design, decision coordination and model evaluation" - but this is not entirely **technical** novelty, rather novelty by design.
>
> Overall, I will keep my positive score of the paper -- but it must be stated that this is highly borderline. If the paper were to be accepted, much of the above must be addressed (including the concerns by other reviewers). Such changes may require a substantial change of the paper, and as such it may require another round of reviews at a subsequent venue.

---

> > ### Author Response · Authors · 2025-12-03
> > **Response to Reviewer wD7m --- Part 3**
> >
> > > Writing issue.
> >
> > We sincerely appreciate the reviewer’s careful reading of our paper. We have conducted careful proofreading to improve the overall clarity and quality of the writing.
> >
> > > The domain-specific keywords still gives me some issues. From my perspective, the authors are too vague on what these keywords actually are. It seems to be a central part of the paper, so I'd expect this to be detailed explicitly. They state that the keywords are case-specific, but does that mean if one were to implement this in practice, you would need experts to annotate each test case for these keywords anyway? Therefore, during this process, wouldn't they spot the issues themselves? What's then the point of the methodology? I am confused on this aspect. Additionally, the authors provide examples in Table 14 - but surely isn't "sepsis" (and many others) a keyword for the first case?
> >
> > We appreciate the reviewer’s thoughtful feedback and and provide clarification below.
> >
> > (1) About what the keywords represent:
> >
> > The domain-specific keywords are directly tied to error types, not to general clinical facts. For example, in MEDEC, the keywords correspond to five error categories: Diagnosis, Management, Treatment, Pharmacotherapy, and Causal Organism. Importantly, factual information such as patient symptoms or pre-existing conditions at admission is not included in the keywords, since we are concerned with the keywords that constitute an error. For instance, in Table 14, “sepsis” is not treated as a keyword because it represents an existing diagnosis at admission (“Mr. <NAME/> was treated for sepsis, present on admission.”) rather than an erroneous or modifiable statement. We added these explanations in Appendix A.10.
> >
> > (2) About how the keywords are obtained:
> >
> > We would like to emphasize that our metric is independent of how the keywords are acquired. For the test sets, we manually labeled the keywords. However, these keywords can also be extracted automatically by comparing the “Error Sentence” and the “Corrected Sentence” provided in the dataset. The keywords are selected from the provided ground-truth. We include the keyword-extraction script in the anonymous link, enabling new datasets to be processed without requiring manual labeling.  The methodology is thus fully flexible, and the implementation choice depends on the researcher’s or institution’s needs. We are happy to run this experiment if the reviewer deems it necessary but we want to focus the paper on the framework.
> >
> > > Regarding the use of open-source versus proprietary models, "we view this as a broader point that is orthogonal to the significance of our proposed methodological framework". I have to disagree on this point. From my anecdotal experience, in the medical domain, the type of models (i.e., open-source versus proprietary) is highly relevant and therefore not independent of the proposed methodological framework, which deeply roots itself within the healthcare domain. It is unlikely to allow sensitive patient data be sent to 3rd parties to access proprietary. It is my opinion that a more thorough benchmarking against open-source models (doesn't have to be medical specialized) is missed.
> >
> > To address this, we have added experiments on Llama-3.3-70B-Instruct, a strong open-source model. The results demonstrate that MedGuards remains consistently effective even when applied to a open-source backbone:
> >
> > | Model                      | Detection Accuracy | Localization Accuracy | Correction ROUGE-1 | Correction BERTScore | Correction BLEURT | KPCS  | P-values  | Improve |
> > |---------------------------|--------------------|------------------------|---------------------|-----------------------|--------------------|-------|-----------|---------|
> > | Llama-3.3-70B-Instruct    | 0.621              | 0.330                  | 0.397               | 0.395                 | 0.401              | 0.360 | 4.23E-03  | –       |
> > | + MedGuards             | 0.684              | 0.576                  | 0.528               | 0.530                 | 0.536              | 0.487 | –         | 33.10% |
> >
> > These results, with statistically significant gains (P value < 0,05), confirm that MedGuards substantially improves safety-critical medical text correction even when using a fully open-source model. We appreciate your suggestion and believe the added experiments strengthen the paper by showing that MedGuards is effective, model-agnostic, and suitable for privacy-preserving healthcare settings.

---

### Official Review · Reviewer_xh1Y · 2025-10-27

**Soundness:** 2
**Presentation:** 2
**Contribution:** 1
**Rating:** 2
**Confidence:** 4

**Summary:**

This paper introduces MEDGUARD, a multi-agent framework designed to enhance the performance of error detection and correction in medical generated text. The authors also address a limitation in existing evaluation metrics, noting that they prioritize general semantic similarity and can assign high scores even when critical medical keywords are omitted. To address this gap, the paper proposes a novel metric, the Keyword-Prioritized Correction Score (KPCS).

**Strengths:**

- The framework offers a resource-effective and time-efficient approach to enhancing model reliability by improving performance without requiring additional fine-tuning of the base LLMs.
- The framework enhances interpretability by generating reasoning traces during each stage.

**Weaknesses:**

- The multi-agent self-consistency mechanism appears flawed as it only triggers arbitration in cases of disagreement. The paper fails to specify a protocol for low-confidence consensus, where both agents agree on a decision but both report very low confidence scores. Relying on such an agreement is potentially risky.
- The justification for using precisely two agents for the detection and localization steps is not provided. An ablation study is necessary to explore how performance varies with the number of agents to justify this specific design choice. As the authors already mentioned, employing multiple agents would be expensive, yet without a proper ablation, it remains unclear whether using all agents provide a justifiable performance gain relative to its cost.
- The validation for the proposed KPCS metric appears insufficient, primarily because the binary nature of its K() function is problematic. As this function assigns a full score of 1 if at least one keyword is present, a correction could receive a high score while still omitting other critical medical keywords.
- The description of the datasets is somewhat brief. While MEDEC is referenced, a more detailed breakdown of the MedErrBench dataset, particularly regarding its construction, validation, and characteristics, would be beneficial. We couldn't find out the source of MedErrBench, which appears to be an in-house benchmark without explicit clarification. Moreover, if the English, Chinese, and Arabic versions are merely translations of the same dataset, they may not adequately capture language-specific clinical nuances, thus limiting the validity of the claimed multiple benchmark evaluation.

**Questions:**

- The current method inputs the entire note, extracts an erroneous sentence, and then uses string similarity to align it with the original text. Did the authors consider a sentence-by-sentence processing approach? This could potentially eliminate string alignment step. Furthermore, the per-sentence reasoning traces from such a process could be directly fed into the correction agent to provide more localized context.
- Regarding the "critical keywords" used in the KPCS metric: how are these defined and extracted? Are these keywords extracted from the reference sentence Sr automatically or are they manually annotated? Does this check require a strict exact string match, or does the mechanism also account for semantic equivalents?
- The citation order appears inconsistent, with some listed chronologically and others in reverse order; it would be preferable to standardize the format throughout the paper.
- The explanation of the self-consistency procedure is somewhat insufficient. While the code suggests that multiple prompts were used for a single model to ensure consistency, this point is not clearly described in the main text and should be elaborated for clarity.
- It would be helpful to clarify how the proposed metric differs from MEDCON [1]. In particular, instead of using a binary (0/1) scale, adopting MEDCON's keyword-level checking mechanism might provide a more informative comparison.
- The value of α in KPCS metric is not specified. Is there an empirically optimal α that shows the highest correlation with human evaluation? Including such details would strengthen the metric's interpretability.
- In line 249, the comma before "of other agents" appears to be a grammatical error or typographical mistake.
- In line 306, and 351, "appendixA.2" and "appendixA.3" should be consistently capitalized. The inconsistent use of lowercase "appendix" should be corrected.
- In line 449, the figure reference seems incorrect; it should likely refer to Figure 1 rather than Figure 3, since Figure 3 presents MedErrBench, not MEDEC.
- In line 465, the figure number is missing and should be specified for clarity.
- In line 736, "and5" lacks a space and should be corrected to "and 5."
----
*[1] Aci-bench: a Novel Ambient Clinical Intelligence Dataset for Benchmarking Automatic Visit Note Generation*

---

> ### Author Response · Authors · 2025-11-23
> **Response to reviewer xh1Y --- Part 1**
>
> Thank you for your constructive feedback! Below is our detailed response.
>
> > The multi-agent self-consistency mechanism appears flawed as it only triggers arbitration in cases of disagreement. The paper fails to specify a protocol for low-confidence consensus, where both agents agree on a decision but both report very low confidence scores. Relying on such an agreement is potentially risky.
>
> We appreciate the reviewer’s concern about low-confidence consensus. That is a good point. But this is not a design flaw and the protocol for low-confidence consensus is not suitable for MedGuards. On the MEDEC dataset, we performed a confidence score analysis using the best performing MedGuards model (based on Doubao-1.5-thinking-pro) focusing on agreement cases. We compared confidence scores between correct and wrong predictions:
>
> | Average confidence score | Detection - agent 1 | Detection - agent 2 |
> |---------------------------|--------------------|--------------------|
> | All case                  | 96.271             | 95.764             |
> | Correct case              | 96.239             | 95.836             |
> | Wrong case                | 96.406             | 95.469             |
>
> We observe that the confidence scores are not directly related to whether a case is correct or incorrect. Thus, confidence alone cannot reliably distinguish correctness, and using a low-confidence threshold to trigger arbitration is not suitable for MedGuards. However, this is an interesting area of future work that also requires investigating calibration of the models.
>
>
> > The justification for using precisely two agents for the detection and localization steps is not provided. An ablation study is necessary to explore how performance varies with the number of agents to justify this specific design choice. As the authors already mentioned, employing multiple agents would be expensive, yet without a proper ablation, it remains unclear whether using all agents provide a justifiable performance gain relative to its cost.
>
> We thank the reviewer for the valuable comment. We have added an ablation study that examines how performance varies with the number of agents. The MedGuards are based on Gemini 2.0 Flash. The new results (now included in Table 11) are summarized below:
>
> | Detection | Localization | Correction | Number of total agents | Detection Accuracy | Localization Accuracy | Correction ROUGE-1 | Correction BERTScore | Correction BLEURT | Average Score |
> |------------|--------------|-------------|-------------------------|--------------------|-----------------------|--------------------|----------------------|-------------------|----------|
> | 1 agent for all task |  |  | 1 | 0.589 | 0.415 | 0.379 | 0.380 | 0.395 | 0.432 |
> | 1 agent for detection and localization |  | 1 agent | 2 | 0.688 | 0.484 | 0.410 | 0.446 | 0.511 | 0.508 |
> | 1 agent | 1 agent | 1 agent | 3 | 0.650 | 0.250 | 0.439 | 0.416 | 0.513 | 0.454 |
> | 3 agent | 1 agent for localization and correction |  | 4 | 0.713 | 0.501 | 0.535 | 0.521 | 0.515 | 0.557 |
> | 3 agent | 3 agent | 1 agent | 7 | 0.713 | 0.609 | 0.537 | 0.549 | 0.549 | 0.591 |
> | 3 agent | 3 agent | 2 agent | 8 | 0.713 | 0.609 | 0.367 | 0.367 | 0.374 | 0.486 |
>
> Our MedGuards (3 agents for detection, 3 agents for localization, and 1 agent for correction) achieves the strongest overall performance compared to the other structure of multi-agent framework.

---

> ### Author Response · Authors · 2025-11-23
> **Response to reviewer xh1Y --- Part 2**
>
> > The validation for the proposed KPCS metric appears insufficient, primarily because the binary nature of its K() function is problematic. As this function assigns a full score of 1 if at least one keyword is present, a correction could receive a high score while still omitting other critical medical keywords.
>
> We thank the reviewer for the valuable comment. We modified it to K() = n/m, where n is the number of keywords in the corrected sentence S_c and m is the number of keywords in S_r. We accordingly modified Section 3.4 and the KPCS in Table 2. To further examine its effect and identify an empirically optimal value, we conducted two additional analyses:
>
> (1) evaluated system performance under different α values (0.2, 0.5, 0.8), and
>
> (2) collected new human evaluation results (Normalized MOS Scores) from an additional clinician.
>
> | Alpha | Gemini 2.0 Flash + MedGuards | GPT-4o- mini  + MedGuards | Doubao-1.5-thinking-pro + MedGuards | Deepseek-v3 + MedGuards | Gemini 2.5 Flash Lite + MedGuards |
> |:------:|:-----------------------------:|:------------------:|:----------------------------------:|:-----------------------:|:---------------------------------:|
> | 0.2 | 0.516 | 0.524 | 0.594 | 0.599 | 0.37 |
> | 0.5 | 0.472 | 0.461 | 0.554 | 0.556 | 0.385 |
> | 0.8 | 0.428 | 0.38 | 0.513 | 0.513 | 0.294 |
>
> | Models                        | Normalized MOS Score |
> |-------------------------------|----------------------|
> | Gemini 2.0 Flash              | 0.425                |
> | Gemini 2.0 Flash + MedGuards  | 0.483                 |
> | GPT-4o-mini                   | 0.383                |
> | GPT-4o-mini + MedGuards       | 0.475                |
>
> The results indicate that α = 0.5 yields KPCS scores most consistent with human evaluation trends. For both Gemini 2.0 Flash and GPT-4o-mini, the model ranking under α = 0.5 best matches the human Normalized MOS results, while α = 0.2 or 0.8 tends to over or under emphasize differences.
>
>
> > The description of the MedErrBench datasets is somewhat brief. We couldn't find out the source of MedErrBench, which appears to be an in-house benchmark without explicit clarification. Moreover, if the English, Chinese, and Arabic versions are merely translations of the same dataset, they may not adequately capture language-specific clinical nuances, thus limiting the validity of the claimed multiple benchmark evaluation.
>
> Thank you for highlighting this ambiguity. We confirm that it is an internal dataset. Importantly, all primary experiments supporting the methodological contributions are performed on the public MEDEC dataset to maintain reproducibility; the internal dataset is used only in Table 3 to provide an additional multilingual applicability check. Since the primary contribution of the paper is the methodology, not the dataset itself, we revised the terminology to refer to this resource as an “internal multilingual dataset” to avoid any confusion.
>
> Please see below the requested details, which now have been added to Appendix A.1: “The internal dataset was constructed to evaluate multilingual medical-error correction and contains 2,506 clinical texts across three languages: English (1,024), Chinese (1,000), and Arabic (482). The English and Chinese subsets were sampled and perturbed from MedQA, while the Arabic subset was derived from MedArabiQ. Each item was designed to cover one of ten clinician-defined medical error categories: diagnosis, management, treatment, pharmacotherapy, causal organism/pathogen, lab/serum value interpretation, physiology, histology, anatomy, and epidemiology. To ensure quality, we applied a two-step human review process: (1) NLP researchers verified fluency and plausibility, and (2) two independent clinicians per language validated medical correctness and clinical realism. No translations were used to preserve language-specific nuances..”
>
> We hope this clarifies the purpose, construction, and role of the internal dataset in our evaluation. We also plan to release an expanded version of this dataset as a public benchmark in future work.

---

> ### Author Response · Authors · 2025-11-23
> **Response to reviewer xh1Y --- Part 3**
>
> > Did the authors consider a sentence-by-sentence processing approach? This could potentially eliminate string alignment step. Furthermore, the per-sentence reasoning traces from such a process could be directly fed into the correction agent to provide more localized context.
>
> We process the entire clinical note to preserve global context, as inter-sentence dependencies are common in clinical narratives. The task is not strictly sequential, identifying errors often requires considering the whole note rather than isolated sentences. Here’s an example:
>
> Erroneous clinical note: “Mr. <NAME/> was Mr. <NAME/> treated for sepsis, present on admission. Pt is a <AGE/> yo M who was admitted with hypoxia, cough and AMS; pt is discovered to have pneumonia and blood cultures from admission grew Strep pneumococcus. Pt presented with WBC 29, tachypneic, with AKI and somnolence/lethargy. Mental status much improved and Fever and WBC coming down on hospital day #2 following IV abx and IVF; an 8 day course of Antifungal Medication at home will be completed.”
>
> Corrected clinical note: “Mr. <NAME/> was Mr. <NAME/> treated for sepsis, present on admission. Pt is a <AGE/> yo M who was admitted with hypoxia, cough and AMS; pt is discovered to have pneumonia and blood cultures from admission grew Strep pneumococcus. Pt presented with WBC 29, tachypneic, with AKI and somnolence/lethargy. Mental status much improved and Fever and WBC coming down on hospital day #2 following IV abx and IVF; an 8 day course of antibiotics at home will be completed.”
>
> If the sentence “an 8 day course of Antifungal Medication at home will be completed” is evaluated separately, it is difficult to determine whether it is correct or not. However, when considering the overall clinical context, the use of an antifungal medication is clearly inappropriate, and the correct treatment should be antibiotics. This example illustrates that analyzing a single sentence without context will lead to incorrect judgments due to the lack of contextual information. While a sentence-level approach could simplify alignment, it risks losing important cross-sentence information in the clinical setting.
>
> > Regarding the "critical keywords" used in the KPCS metric: how are these defined and extracted? Are these keywords extracted from the reference sentence Sr automatically or are they manually annotated? Does this check require a strict exact string match, or does the mechanism also account for semantic equivalents?
>
> (1） The critical keywords refer to key clinical concepts, such as diagnoses, medications, therapeutic actions, or essential findings, that are central to the clinical meaning of each note and particularly relevant to potential errors. These entities were manually annotated by our team across all samples. To clarify this in the revised version, we have included  a new Table (now in Table 14) in the updated manuscript illustrating representative examples of critical clinical entities.
> For instance, in the following sample:
>
> Erroneous clinical note: “Mr. <NAME/> was Mr. <NAME/> treated for sepsis, present on admission. Pt is a <AGE/> yo M who was admitted with hypoxia, cough and AMS; pt is discovered to have pneumonia and blood cultures from admission grew Strep pneumococcus. Pt presented with WBC 29, tachypneic, with AKI and somnolence/lethargy. Mental status much improved and Fever and WBC coming down on hospital day #2 following IV abx and IVF; an 8 day course of Antifungal Medication at home will be completed.”
>
> Corrected clinical note: “Mr. <NAME/> was Mr. <NAME/> treated for sepsis, present on admission. Pt is a <AGE/> yo M who was admitted with hypoxia, cough and AMS; pt is discovered to have pneumonia and blood cultures from admission grew Strep pneumococcus. Pt presented with WBC 29, tachypneic, with AKI and somnolence/lethargy. Mental status much improved and Fever and WBC coming down on hospital day #2 following IV abx and IVF; an 8 day course of antibiotics at home will be completed.”
>
> Here, the critical clinical entity is “antibiotics”, which directly determines the correctness of the therapeutic action based on the overall context of the note, since antifungal treatment would be clinically inappropriate for bacterial pneumonia.
>
> （2）We currently use exact string matching for the first component of the KPCS score. Implementing a more flexible alternative would require a principled approach to synonym mapping, likely through integrating medical ontologies or structured clinical knowledge bases, since generic semantic similarity measures would not provide sufficient precision for clinical keywords. We will add this point as a future direction in the discussion section.
>
> It is also important to note that the second term of the KPCS score, M(⋅), already captures broader semantic similarity (Step 2). Thus, the overall metric balances both precise keyword verification and semantic and lexical similarity, preserving the strengths of both perspectives.

---

> ### Author Response · Authors · 2025-11-23
> **Response to reviewer xh1Y --- Part 4**
>
> > The citation order appears inconsistent, with some listed chronologically and others in reverse order; it would be preferable to standardize the format throughout the paper.
>
> Thank you for pointing this out. We have adjusted the citation order in the manuscript.
>
> > The explanation of the self-consistency procedure is somewhat insufficient. While the code suggests that multiple prompts were used for a single model to ensure consistency, this point is not clearly described in the main text and should be elaborated for clarity.
>
> Thank you for the suggestion. We added the explanation in the experimental setting with the following sentence: “In our framework, self-consistency during the detection and localization stages is achieved by employing multiple models with distinct prompts, rather than relying on repeated prompting of a single model. Each model separately generates its output, and their results are compared to reach a consistent decision through arbitration if disagreement occurs.”
>
>
> > It would be helpful to clarify how the proposed metric differs from MEDCON [1].
>
> Although both MEDCON and our proposed KPCS leverage domain-specific medical terms, their goals and mechanisms differ substantially. MEDCON measures concept-level overlap between UMLS sets in generated and reference notes, thus focusing purely on concept coverage. In contrast, KPCS is designed for sentence-level error correction, where both keyword retention and overall linguistic quality matter.
>
> Specifically, KPCS explicitly separates medical keywords from other words and computes two components: (1) a keyword-presence score assessing whether critical medical terms are preserved, and (2) a semantic similarity score evaluating the overall content. These two parts are then weighted by a tunable factor α, allowing fine control between preciseness and overall correctness. MEDCON lacks such a weighting mechanism and does not differentiate between keyword and non-keyword content, making KPCS more sensitive to clinically important corrections.
>
> > The value of α in KPCS metric is not specified. Is there an empirically optimal α that shows the highest correlation with human evaluation? Including such details would strengthen the metric's interpretability.
>
> We appreciate the reviewer’s comment regarding the selection of α in the KPCS metric. In our experiments, α was set to 0.5. To further examine its effect and identify an empirically optimal value, we conducted two additional analyses:
>
> (1) evaluated system performance under different α values (0.2, 0.5, 0.8), and
>
> (2) collected new human evaluation results (Normalized MOS Scores) from an additional clinician.
>
> | Alpha | Gemini 2.0 Flash + MedGuards | GPT-4o- mini  + MedGuards | Doubao-1.5-thinking-pro + MedGuards | Deepseek-v3 + MedGuards | Gemini 2.5 Flash Lite + MedGuards |
> |:------:|:-----------------------------:|:------------------:|:----------------------------------:|:-----------------------:|:---------------------------------:|
> | 0.2 | 0.516 | 0.524 | 0.594 | 0.599 | 0.37 |
> | 0.5 | 0.472 | 0.461 | 0.554 | 0.556 | 0.385 |
> | 0.8 | 0.428 | 0.38 | 0.513 | 0.513 | 0.294 |
>
> | Models                        | Normalized MOS Score |
> |-------------------------------|----------------------|
> | Gemini 2.0 Flash              | 0.425                |
> | Gemini 2.0 Flash + MedGuards  | 0.483                 |
> | GPT-4o-mini                   | 0.383                |
> | GPT-4o-mini + MedGuards       | 0.475                |
>
> The results indicate that α = 0.5 yields KPCS scores most consistent with human evaluation trends. For both Gemini 2.0 Flash and GPT-4o-mini, the model ranking under α = 0.5 best matches the human Normalized MOS results, while α = 0.2 or 0.8 tends to over or under emphasize differences.
>
>
> > Capitalization, spacing, punctuation, figure numbering
>
> Thank you for pointing these out. These issues have been corrected in the revised version.
>
> We kindly ask you to evaluate our submission in light of the above feedback and improvements made to our manuscript. Thank you for your time and effort.

---

> > ### Comment · Reviewer_xh1Y · 2025-11-25
> > **Official Comment by Reviewer xh1Y**
> >
> > Thank you for the detailed rebuttal and additional experimental results. I increased the contribution score. However, I still have several remaining concerns:
> >
> > ---
> >
> > - You compare performance using MEDEC and an internal benchmark (formerly MedErrBench), but both datasets are constructed based on MedQA. This makes a concern that the framework might be overfitted to the MedQA format, and its generalization performance on real-world clinical data, which contains a lot of noise, is worrisome.
> > - The authors stated that the critical keywords were manually annotated by team members. This implies that human resources must be invested every time this metric is applied to a new dataset or domain, which could be a constraint on the scalability of the metric.
> > - The background for asking about the difference between KPCS and MEDCON in the previous review was that the K() term in KPCS seemed to have a similar purpose to MEDCON. If the K() term, which utilizes critical keywords requiring manual annotation as explained, is replaced with the automated MEDCON, it seems it could solve the aforementioned manual annotation problem while also reflecting Semantic similarity that the K() term misses. Furthermore, since the K() term can be replaced by the more powerful MEDCON which serves a similar purpose and the M() term utilizes existing metrics, it seems that the novelty of KPCS is somewhat limited.
> > - Regarding the concern about "Low-confidence consensus," you answered that the average confidence scores for correct/wrong cases are similar. Instead of the overall average, could you provide the specific magnitude of score change (performance improvement) when the Arbitration agent intervenes for low-confidence sentences?
> > - In Table 11, it would be helpful for analysis if the trend of accuracy changes when the number of Base agents is increased within each stage is also presented.
> > - In Equation (7), the condition is specified as "if S_c contains more than one keyword". Confirmation is needed on whether the intention is to award points only when there are 2 or more keywords. If the intention was "1 or more" as in the original manuscript, there is no need to divide it with a conditional statement; simply expressing it as n/m would naturally result in 0 when n = 0, so the formula can be simplified.
> > - Table 6: Since the scores are normalized, this should be explicitly stated in the caption.
> > - Table 13: "Correction Rate" and "Corrected Cases" seem more appropriate to be labeled as "Correctness Rate" and "Correct Cases" based on the context.
> > - Line 277: The Appendix reference needs to be corrected from A.9 to A.10.
> > - Line 777: A period (.) is missing at the end of the sentence.
> > - Line 929: The spacing after "in Table 6" needs correction.
> >
> > ---
> >
> > Following those concerns, I have decided to keep my overall scores. Thank you again for your detailed rebuttal.

---

> > > ### Author Response · Authors · 2025-11-28
> > > **Further response to reviewer xh1Y --- Part 5**
> > >
> > > > You compare performance using MEDEC and an internal benchmark (formerly MedErrBench), but both datasets are constructed based on MedQA. This makes a concern that the framework might be overfitted to the MedQA format, and its generalization performance on real-world clinical data, which contains a lot of noise, is worrisome.
> > >
> > > We understand the reviewer’s concern. However, our framework does not process the data in a multiple-choice or QA format. Instead, each case is provided to the model as a single clinical note with embedded errors for detection, localization and correction, which differs substantially from the question–answer pairs used in MedQA. This is also why MEDEC has become a popular choice for benchmarking medical error detection tasks.
> > >
> > > Regarding the use of real-world clinical data, we fully agree that this is an important next step. However, annotated real-world datasets for medical error detection and correction are currently unavailable, as this research area is still in its early stages and manual annotation would require substantial clinical expertise and cost. Therefore, as in prior work in this domain, we rely on a controlled and well-defined benchmark dataset to establish a proof of concept. We have noted this point in the discussion section for future work.
> > >
> > > >The authors stated that the critical keywords were manually annotated by team members. This implies that human resources must be invested every time this metric is applied to a new dataset or domain, which could be a constraint on the scalability of the metric.
> > >
> > > We would like to clarify that the metric itself is independent of how the keywords are obtained. For MEDEC and internal benchmark, we manually labeled the keywords. However, these keywords can also be extracted automatically by comparing the “Error Sentence” and the “Corrected Sentence” provided in the dataset. The keywords are selected from the provided ground-truth. We included the rule-based keyword-extraction script in the anonymous link, enabling new datasets to be processed without requiring manual labeling.
> > >
> > > The key idea of our evaluation metric KPCS is to distinguish between keywords and non-keywords within the segment of the corrected text that’s supposed to have contained the error when computing the score, while the process of obtaining the keywords is fully flexible. This is similar to widely used evaluation metrics such as Dice, which measures segmentation quality regardless of whether the ground truth annotations are human- or machine-generated. We need such standardized metrics for error correction tasks to understand how well the model performs in the context of specific corrections, beyond the overall semantics of the generated text.
> > >
> > > >The background for asking about the difference between KPCS and MEDCON in the previous review was that the K() term in KPCS seemed to have a similar purpose to MEDCON. If the K() term, which utilizes critical keywords requiring manual annotation as explained, is replaced with the automated MEDCON, it seems it could solve the aforementioned manual annotation problem while also reflecting Semantic similarity that the K() term misses. Furthermore, since the K() term can be replaced by the more powerful MEDCON which serves a similar purpose and the M() term utilizes existing metrics, it seems that the novelty of KPCS is somewhat limited.
> > >
> > > We would like to clarify that KPCS and MEDCON serve very different purposes.
> > >
> > > KPCS is designed to distinguish between keyword and non-keyword content when comparing the (i) the generated “corrections” and (ii) the ground truth text, allowing fine control between preciseness and overall correctness via a tunable parameter alpha α.
> > >
> > > In contrast, MEDCON captures UMLS concepts explicitly mentioned in the overall transcript “to gauge the accuracy and consistency of clinical concepts” [1]. Hence it is less concerned with the specific corrected parts of the text as in KPCS.
> > > An interesting direction of future work is to consider a hybrid metric that also considers UMLS concepts as an additional feature, but this is beyond the current objective of our proposed metric.
> > >
> > > [1] Yim, Wen-wai, et al. "Aci-bench: a novel ambient clinical intelligence dataset for benchmarking automatic visit note generation." Scientific data 10.1 (2023): 586.
> > >
> > > > In Equation (7), the condition is specified as "if S_c contains more than one keyword". Confirmation is needed on whether the intention is to award points only when there are 2 or more keywords. If the intention was "1 or more" as in the original manuscript, there is no need to divide it with a conditional statement; simply expressing it as n/m would naturally result in 0 when n = 0, so the formula can be simplified.
> > >
> > > Thank you for your suggestion. We confirm that points are awarded when one or more keywords are contained in Sc, and we have revised the equation accordingly.

---

> > > ### Author Response · Authors · 2025-11-28
> > > **Further response to reviewer xh1Y --- Part 6**
> > >
> > > > Regarding the concern about "Low-confidence consensus," you answered that the average confidence scores for correct/wrong cases are similar. Instead of the overall average, could you provide the specific magnitude of score change (performance improvement) when the Arbitration agent intervenes for low-confidence sentences?
> > >
> > > We conducted a further analysis of the model’s confidence scores. Among the agreement cases, the numbers of instances where both agents’ confidence scores are simultaneously:
> > >
> > > <90 is 18 (1.95%)
> > >
> > > <85 is 2 (0.22%)
> > >
> > > <80 is 0 (0%)
> > >
> > > In cases where both agents have confidence scores below 85, the two agents agree 100% of the time, and all predictions are correct. Therefore, in this setting, the arbitration agent does not need to intervene for the low-confidence samples.
> > > In cases where both agents have confidence scores below 90, the agents agree in (17/18) 94.4% of instances, and all of these predictions are correct. In the remaining (1/18) 5.6% of instances, the two agents disagree. After entering the Arbitration stage, the system makes an incorrect prediction. Hence, under this setting, the arbitration agent did not need to intervene for two low-confidence samples either. Therefore, under the current MedGuards framework, this mechanism would not affect model outcomes.
> > >
> > > Furthermore, we would also like to note that while including the confidence scores did lead to improvements in performance during our ablations mostly for the error detection step (Table 4), further work is needed to investigate these findings in the context of LLM agent calibration, which is an active area of research [2].
> > >
> > > [2] Geng, Jiahui, et al. "A survey of confidence estimation and calibration in large language models." Proceedings of the 2024 Conference of the North American Chapter of the Association for Computational Linguistics: Human Language Technologies (Volume 1: Long Papers). 2024.
> > >
> > > > In Table 11, it would be helpful for analysis if the trend of accuracy changes when the number of Base agents is increased within each stage is also presented.
> > >
> > > We appreciate the reviewer’s suggestion. We further analyzed the performance trend when increasing the number of base agents within each stage. Overall, the detection and localization stages tend to show improved performance, although this trend is not strictly monotonic. For example, using one agent for each of the three stages performs worse than sharing a single agent between detection and localization. Compared to the single-agent baseline, the two-agent setting yields a 17.6% improvement in the overall average score, and expanding to four agents further increases the gain to 28.9%. When both detection and localization adopt three agents with arbitration, the performance reaches its best improvement of 36.8%. In contrast, the correction stage exhibits an opposite trend: increasing the number of correction agents leads to a drop in generation quality, with performance falling to only a 12.5% improvement when two correction agents are used. This is likely because correction is a generation task, where multiple agents may produce different valid outputs that are difficult to unify, making consensus-based arbitration challenging. We add this explanation in Appendix A.7.
> > >
> > > > Caption, name, appendix index and space.
> > >
> > > Thank you for pointing these out. These issues have been corrected in the revised version.
> > >
> > > We kindly ask you to evaluate our submission in light of the overall improvements made to our manuscript. We believe we have addressed all major concerns, which has significantly improved the clarity and reliability of our manuscript. We are happy to address any further questions/ comments.

---

### Official Review · Reviewer_Zuwj · 2025-10-30

**Soundness:** 3
**Presentation:** 2
**Contribution:** 3
**Rating:** 6
**Confidence:** 3

**Summary:**

This paper proposes a framework (called MedGuards) that treats medical error detection and correction as a multi-agent in-context learning task. additionally, the paper introduces an evaluation metric (called KPCS) that assigns greater weights to critical clinical entities.

**Strengths:**

1. The paper's methodology section is very clear and contains lots of relevant information that might be useful to understand the framework being proposed in this work.

2. being able to validate the proposed framework (medguards) using human evals is a plus point.

3. The work builds on theoretical framework and then validate it using experimental setup across multiple datasets

4. This work is useful in medical setup where medical errors can be automatically deducted. hence there is a real life implication

**Weaknesses:**

1. The paper has shared the anonymous github link, however it would be useful to have the prompts in appendix for quick reference

2. The paper does not mention the false positives or negatives by the agents at each stage. It would be interesting to see (qualitatively) if there is any patterns where a specific llm fails (and perhaps other llm did not fail).

**Questions:**

1. Can you clarify whether each human evaluator rated all cases or only a subset? If each evaluator rated all the cases, I believe it might be a good idea to include inter-rater agreement to show how much humans agree.

2. The table 1 title does not specify which dataset results are being shown? It took me a while after reading to infer but it would be helpful to have that information right in the title itself (similar to table 2 wher you mentioned the dataset name)

3. What do you mean by "best MedGuards-enhanced model" in table 1's caption? I think some clarity is needed here.

4. I understood that one single LLM is being utilized in three agents. However, would it be possible to see how the mixture of models would change the performance? For instance you can create a permutation and combination of all models for all the stages (detection, localization and correction). I do see Table 7 doing that but it is (a) limitated to one dataset (b) not showing all possible combinations.

---

> ### Author Response · Authors · 2025-11-23
> **Response to Reviewer Zuwj --- Part 1**
>
> Thank you for your constructive feedback! Below is our detailed response.
>
> > The paper has shared the anonymous github link, however it would be useful to have the prompts in appendix for quick reference
>
> Thank you for the suggestion. We added the prompts to new Appendix Section A.11 in Table 15.
>
>
> > The paper does not mention the false positives or negatives by the agents at each stage. It would be interesting to see (qualitatively) if there is any patterns where a specific llm fails.
>
>
> We agree that this is an interesting analysis. We have further analyzed the qualitative behavior of the agents at each stage on MEDEC dataset:
>
>
> | **Doubao-1.5-Thinking-Pro + MedGuards** | **Detection - Agreement** | **Detection - Arbitration** | **Localization - Agreement** | **Localization - Arbitration** |
> |----------------|---------------------------|-----------------------------|-------------------------------|--------------------------------|
> | **Case Number (percentage)** | 818 (88.43%) | 107 (11.57%) | 401 (86.61%) | 61 (13.20%) |
> | **Corrected percentage (corrected case num)** | 80.44% (658) | 46.73% (50) | 71.07% (285) | 49.18% (30) |
> | **Wrong percentage (wrong case num)** | 19.56% (160) | 53.27% (57) | 28.93% (116) | 50.82% (31) |
>
>
>
> | **GPT-4o-mini + MedGuards** | **Detection - Agreement** | **Detection - Arbitration** | **Localization - Agreement** | **Localization - Arbitration** |
> |----------------|---------------------------|-----------------------------|-------------------------------|--------------------------------|
> | **Case Number (percentage)** | 725 (78.4%) | 200 (21.6%) | 274 (48.4%) | 292 (51.6%) |
> | **Corrected percentage (corrected case num)** | 69.24% (502) | 48.5% (97) | 47.4% (130) | 33.2% (97) |
> | **Wrong percentage (wrong case num)** | 30.76% (223) | 51.5% (103) | 52.6% (144) | 66.8% (195) |
>
> The results show the lower-performing GPT-4o-mini + MedGuards model shows a higher rate of agent disagreement in both detection and localization stages, thus entering the arbitration stage more frequently. Second, for both models, cases entering arbitration exhibit lower accuracy. The arbitration process still corrects about half of such cases, indicating partial complementarity between agents. Moreover, localization tasks generally show higher disagreement rates than detection, suggesting that fine-grained spatial reasoning remains more challenging. These analyses provide qualitative insights into the failure patterns of each LLM and demonstrate the robustness and diagnostic value of the MedGuards design. We added these experiments in the Appendix A.9.
>
> > Can you clarify whether each human evaluator rated all cases or only a subset? If each evaluator rated all the cases, I believe it might be a good idea to include inter-rater agreement to show how much humans agree.
>
> Each human evaluator rated a subset of the samples rather than all cases. As described in Appendix A.3, “For each set of results, we randomly sampled 20 cases, resulting in a total of 20 × 4 = 80 cases for evaluation.”
>
> To further improve the reliability of the human evaluation, during the rebuttal period we invited an additional experienced clinician to independently assess the same set of cases. The newly collected ratings showed a 57.5% agreement, reflecting the consistency among clinicians while still allowing for reasonable variability. The agreement was computed as the proportion of cases where both clinicians assigned the same score to the corresponding model outputs for a given case. Since each clinician rated cases based on their own medical experience and judgment, some variation across ratings is expected and aligns with real-world practice. We have added this in Appendix Section A.3.
>
> > The table 1 title does not specify which dataset results. The meaning of "best MedGuards-enhanced model" in table 1's caption.
>
> Thank you for pointing this out. We added the MEDEC dataset in the caption. The term “best MedGuards-enhanced model” referred to the MedGuards framework built on top of the base LLM that achieved the highest performance among all MedGuards variants evaluated in Table 2. To avoid any confusion, we amended the Table caption for better clarity: “the performance of our framework MedGuards, which is the best variant with Doubao-1.5 thinking pro as the base LLM”.

---

> ### Author Response · Authors · 2025-11-23
> **Response to Reviewer Zuwj --- Part 2**
>
> > I understood that one single LLM is being utilized in three agents. However, would it be possible to see how the mixture of models would change the performance? For instance you can create a permutation and combination of all models for all the stages. I do see Table 7 doing that but it is (a) limitated to one dataset (b) not showing all possible combinations.
>
> We appreciate the reviewer’s suggestion to explore how different mixtures of models affect performance. Our framework consists of three stages (detection, localization, and correction), involving 7 agents in total, each of which can be instantiated with one of five LLMs. If each agent is allowed to select an LLM independently, this would result in 5^7 =78125 possible combinations. Even under the simplified assumption that all agents within the same stage share the same model, there would still be  5×5×5−3=122 possible combinations. Exhaustively evaluating all these combinations is infeasible.
>
> To address this concern, we conducted additional experiments covering representative configurations across stages, including: (a) variations where different models are mixed within a stage, and (b) combinations where stages use distinct models. The results are summarized below:
>
> | Detection Agent | Localization Agent | Correction Agent | Detection Accuracy | Localization Accuracy | Correction ROUGE-1 | Correction BERTScore | Correction BLEURT | Average Score |
> |------------------|--------------------|------------------|--------------------|-----------------------|--------------------|----------------------|-------------------|----------------|
> | 3 Gpt-4o-mini | 3 Gpt-4o-mini | Gpt-4o-mini | 0.695 | 0.535 | 0.571 | 0.581 | 0.574 | 0.591 |
> | 2 Gpt-4o-mini + 1 Gemini 2.0 Flash | 3 Gpt-4o-mini | Gpt-4o-mini | 0.652 | 0.563 | 0.506 | 0.509 | 0.504 | 0.547 |
> | (1 Gpt-4o-mini + 1 Gemini 2.0 Flash) + 1 Gemini 2.0 Flash | 3 Gpt-4o-mini | Gpt-4o-mini | 0.674 | 0.576 | 0.517 | 0.521 | 0.515 | 0.561 |
> | 3 Gpt-4o-mini | 2 Gpt-4o-mini + 1 Gemini 2.0 Flash | Gpt-4o-mini | 0.695 | 0.580 | 0.518 | 0.521 | 0.512 | 0.565 |
> | 3 Gpt-4o-mini | (1 Gpt-4o-mini + Gemini 2.0 Flash) + 1 Gemini 2.0 Flash | Gpt-4o-mini | 0.695 | 0.638 | 0.577 | 0.577 | 0.565 | 0.610 |
>
> | Detection Agent | Localization Agent | Correction Agent | Detection Accuracy | Localization Accuracy | Correction ROUGE-1 | Correction BERTScore | Correction BLEURT | Average Score |
> |------------------|--------------------|------------------|--------------------|-----------------------|--------------------|----------------------|-------------------|----------------|
> | Gpt-4o-mini | Gpt-4o-mini | Gpt-4o-mini | 0.695 | 0.535 | 0.571 | 0.581 | 0.574 | 0.591 |
> | Gpt-4o-mini | Gemini 2.0 Flash | Gpt-4o-mini | 0.695 | 0.583 | 0.543 | 0.543 | 0.533 | 0.579 |
> | Gpt-4o-mini | Gpt-4o-mini | Gemini 2.0 Flash | 0.695 | 0.535 | 0.563 | 0.580 | 0.502 | 0.579 |
> | Gemini 2.0 Flash | Gpt-4o-mini | Gpt-4o-mini | 0.713 | 0.484 | 0.547 | 0.536 | 0.573 | 0.567 |
>
> We used Gpt-4o-mini as the base model for MedGuards. Most other model mixtures lead to slightly lower performance. However, we also observe an interesting case: when Gemini 2.0 Flash is introduced as an additional agent in the localization stage (serving as an agreement agent), the localization and correction performance increase. This suggests that combining complementary LLMs can bring marginal gains in specific sub-tasks but requires further investigation.
>
> We kindly ask you to evaluate our submission in light of the above feedback and improvements made to our manuscript. Thank you for your time and effort.

---

### Official Review · Reviewer_u4tR · 2025-10-31

**Soundness:** 3
**Presentation:** 3
**Contribution:** 3
**Rating:** 4
**Confidence:** 3

**Summary:**

The paper introduces MedGuards, a novel multi-agent system designed to enhance reliability in medical error detection and correction (MEDEC) within LLM-generated clinical text, addressing the limitations of single-model approaches in high-stakes domains. MedGuards employs a CoT-guided decomposition into detection, localization, and correction stages, utilizing specialized agents coordinated by a confidence-guided ICL-based arbitration mechanism that resolves disagreements using reasoning traces. To complement the framework, the authors propose the Keyword-Prioritized Correction Score (KPCS), a domain-specific evaluation metric that emphasizes the fidelity of critical clinical entities. Extensive experiments across four multilingual medical datasets demonstrate robust and significant performance gains over strong LLM baselines and state-of-the-art MEDEC methods, validating the system's effectiveness and clinical safety orientation.

**Strengths:**

- The paper introduces a highly original multi-agent self-consistency framework using ICL-based arbitration specifically designed for the safety-critical task of medical error correction.
- Extensive empirical evaluation across four diverse and multilingual datasets consistently shows significant improvements over strong LLM baselines and specialized prior systems.
- The proposed Keyword-Prioritized Correction Score (KPCS) is a meaningful methodological contribution that successfully aligns evaluation with clinical safety requirements by prioritizing critical entities.
- The methodology is presented clearly, grounding the system design in established principles like CoT decomposition, which contributes to the interpretability and robustness of the pipeline.

**Weaknesses:**

- The reliance on a multi-agent architecture necessitates significant computational overhead and latency, which may hinder its practical deployment speed in time-sensitive clinical settings.
- The paper does not clearly define the source or methodology for extracting the "critical clinical entities" used in the Keyword-Prioritized Correction Score (KPCS).
- The error localization step heavily utilizes simple character-level string similarity (LCS/SequenceMatcher) to align the predicted erroneous sentence with the original text, which may fail for purely semantic errors or paraphrasing.
- The ablation study presented in Table 4 is incomplete as it focuses only on ICL components and omits ablating the core multi-agent structure versus a single high-performing agent pipeline.
- The system's robustness is predicated on the LLM backbones reliably generating structured outputs, including specific XML-like tags for confidence and reasoning, which can be brittle.
- While detection and localization use arbitration, the final error correction stage relies on a single agent prediction, potentially reducing the robust error mitigation intended by the overall framework.
- The parameter $\alpha$ used in the KPCS weighting is described as tunable, but the paper does not specify which value was used for the reported results, impacting reproducibility and metric context.

**Questions:**

- Could the authors provide details on the specific criteria and process used to identify and extract the "critical keywords" required for calculating the KPCS score across the diverse datasets?
- Given that the multi-agent system inherently increases the number of API calls, what is the measured increase in inference latency and computational cost relative to a high-performing single-LLM CoT baseline?
- The final weighted KPCS score depends on the hyperparameter $\alpha$; what value of $\alpha$ was used for the results presented in Tables 2 and 3, and how sensitive is the model ranking to changes in this weighting?

---

> ### Author Response · Authors · 2025-11-23
> **Response to reviewer u4tR --- Part 1**
>
> Thank you for your constructive feedback! Below is our detailed response.
>
> > What is the measured increase in inference latency and computational cost relative to a high-performing single-LLM CoT baseline?
> We thank the reviewer for the valuable comment. To quantitatively assess this concern, we conducted additional experiments to measure the average inference time per clinical notes, average input and output tokens across all agents:
>
> | Models          | Latency (s) | Input tokens (N) | Output tokens (N) | Performance improvement (%) |
> |-----------------|------------------:|----------------------|-----------------------|--------------------:|
> | Gemini 2.0 Flash | 1.61 | 1927 | 48 |  |
> | + MedGuards | 6.99 | Detection: 1047 / Localization: 1127 / Correction: 1135 | Detection: 1159 / Localization: 666 / Correction: 623 | 96.10% |
>
> To translate token usage into computational cost, we used Gemini 2.0 Flash’s public pricing:
>
> | Models           | Input ($/1M tokens) | Output ($/1M tokens) | Context caching ($/1M tokens) |
> |------------------|--------------------:|---------------------:|------------------------------:|
> | Gemini 2.0 Flash | $0.10               | $0.40                | $0.025                        |
>
> While our proposed multi-agent framework MedGuards introduces an increase in latency due to multiple specialized reasoning steps and API calls, the accuracy improvement is crucial in time-sensitive clinical scenarios where decision reliability outweighs minor latency increases. In practical deployment, techniques such as agent parallelization and context caching can further mitigate latency without compromising performance. We included those results in Appendix A.6.
>
>
> > The paper does not clearly define the source or methodology for extracting the "critical clinical entities" used in the Keyword-Prioritized Correction Score (KPCS).
>
> We thank the reviewer for pointing out this concern. The critical clinical entities refer to key clinical concepts, such as diagnoses, medications, therapeutic actions, or essential findings, that are central to the clinical meaning of each note and particularly relevant to potential errors. These entities were manually annotated and then verified by the clinicians across all samples. To clarify this in the revised version, we have included a new Table (Appendix A.10 Table 14) in the updated manuscript illustrating representative examples of critical clinical entities. For instance, in the following sample:
>
> Erroneous clinical note: “Mr. <NAME/> was Mr. <NAME/> treated for sepsis, present on admission. Pt is a <AGE/> yo M who was admitted with hypoxia, cough and AMS; pt is discovered to have pneumonia and blood cultures from admission grew Strep pneumococcus. Pt presented with WBC 29, tachypneic, with AKI and somnolence/lethargy. Mental status much improved and Fever and WBC coming down on hospital day #2 following IV abx and IVF; an 8 day course of Antifungal Medication at home will be completed.”
>
> Corrected clinical note: “Mr. <NAME/> was Mr. <NAME/> treated for sepsis, present on admission. Pt is a <AGE/> yo M who was admitted with hypoxia, cough and AMS; pt is discovered to have pneumonia and blood cultures from admission grew Strep pneumococcus. Pt presented with WBC 29, tachypneic, with AKI and somnolence/lethargy. Mental status much improved and Fever and WBC coming down on hospital day #2 following IV abx and IVF; an 8 day course of antibiotics at home will be completed.”
>
> Here, the critical clinical entity is “antibiotics”, which directly determines the correctness of the therapeutic action based on the overall context of the note, since antifungal treatment would be clinically inappropriate for bacterial pneumonia.

---

> ### Author Response · Authors · 2025-11-23
> **Response to reviewer u4tR --- Part 2**
>
> > The error localization step heavily utilizes simple character-level string similarity (LCS/SequenceMatcher) to align the predicted erroneous sentence with the original text, which may fail for purely semantic errors or paraphrasing.
>
> The error localization task in our framework is intentionally designed to be simple and straightforward. Below is our reasoning with the exact steps of the process:
>
> (1) The main objective here for the LLM is to locate the sentence that contains the error by generating that sentence. We decided to go for this approach because our exploratory analysis showed that asking the model to generate a specific ID led to frequent mistakes so we wanted this to be as exact as possible.
>
> (2) Hence, in the prompt, we explicitly instruct the model to generate exactly the same sentence from the original clinical notes that contain the error.
>
> (3) After generation, we use the SequenceMatcher algorithm to compute character-level similarity between the generated sentence and each sentence in the original clinical notes, selecting the one with the highest score based on the computed similarity ratio. We adopted character-level similarity because the goal of localization is to identify the exact sentence from the original clinical notes, not a paraphrased or semantically similar one, which could be possible within a long medical note and LLMs may occasionally produce minor lexical variations or hallucinations.
>
> (4) This method provides a robust and deterministic way to align the generated output with the true source. Incorporating semantic similarity would, in contrast, risk mismatching sentences that differ in form but not in meaning, which is undesirable for this task.
>
> We included this explanation in Section 3.2 of the revised manuscript.
>
>
>
> > The ablation study presented in Table 4 is incomplete as it focuses only on ICL components and omits ablating the core multi-agent structure versus a single high-performing agent pipeline.
>
> We thank the reviewer for the valuable comment. We would like to clarify that Table 2 and Table 3 in our submission already include the comparison between the core multi-agent structure and a single high-performing agent pipeline.
>
> In addition, we have added an ablation study that further examines different structures of multi-agent framework. The MedGuards are based on Gemini 2.0 Flash. The new results (now included in Table11) are summarized below:
>
> | Detection | Localization | Correction | Number of total agents | Detection Accuracy | Localization Accuracy | Correction ROUGE-1 | Correction BERTScore | Correction BLEURT | Average Score |
> |------------|--------------|-------------|-------------------------|--------------------|-----------------------|--------------------|----------------------|-------------------|----------|
> | 1 agent for all task |  |  | 1 | 0.589 | 0.415 | 0.379 | 0.380 | 0.395 | 0.432 |
> | 1 agent for detection and localization |  | 1 agent | 2 | 0.688 | 0.484 | 0.410 | 0.446 | 0.511 | 0.508 |
> | 1 agent | 1 agent | 1 agent | 3 | 0.650 | 0.250 | 0.439 | 0.416 | 0.513 | 0.454 |
> | 3 agent | 1 agent for localization and correction |  | 4 | 0.713 | 0.501 | 0.535 | 0.521 | 0.515 | 0.557 |
> | 3 agent | 3 agent | 1 agent | 7 | 0.713 | 0.609 | 0.537 | 0.549 | 0.549 | 0.591 |
> | 3 agent | 3 agent | 2 agent | 8 | 0.713 | 0.609 | 0.367 | 0.367 | 0.374 | 0.486 |
>
> Our MedGuards (3 agents for detection, 3 agents for localization, and 1 agent for correction) achieves the strongest overall performance compared to the single-agent baseline and other structure of multi-agent framework. Notably, adding more correction agents (8-agent setting) reduces generation quality, suggesting that our proposed structure provides the most effective balance between collaboration and efficiency.

---

> ### Author Response · Authors · 2025-11-23
> **Response to reviewer u4tR --- Part 3**
>
> > The system's robustness is predicated on the LLM backbones reliably generating structured outputs, including specific XML-like tags for confidence and reasoning, which can be brittle.
>
> Structured outputs with XML-like tags are a widely adopted and well-established practice in LLM-based systems [1, 2, 3]. We observed that it provides high stability in our setup as well. In practice, malformed outputs are rare (<1% of generations)  as the prompts explicitly enforce this format and the generated outputs are automatically validated by simple regex-based checks. Any malformed outputs are easily detected and corrected through lightweight post-processing before further analysis, ensuring overall robustness of the system. We added a few lines in the discussion section highlighting this challenge relating to system robustness and areas of future work.
>
> [1] Montenegro, Larissa, Luis M. Gomes, and José M. Machado. "What We Know About the Role of Large Language Models for Medical Synthetic Dataset Generation." AI 6, no. 6 (2025): 109.
>
> [2] Shen, Zhengyuan, Darren Yow-Bang Wang, Soumya Smruti Mishra, Zhichao Xu, Yifei Teng, and Haibo Ding. "SLOT: Structuring the Output of Large Language Models." In Proceedings of the 2025 Conference on Empirical Methods in Natural Language Processing: Industry Track, pp. 472-491. 2025.
>
> [3] Claude Docs. Use XML tags to structure prompts and outputs. Claude Documentation. Retrieved from https://docs.claude.ai/docs/use-xml-tags.
>
> > While detection and localization use arbitration, the final error correction stage relies on a single agent prediction, potentially reducing the robust error mitigation intended by the overall framework.
>
> We intentionally use a single-agent design for the error correction stage to balance computational cost and latency, correction is less ambiguous once detection/localization have been resolved. Unlike detection or localization, correction is a generation task, where multiple agents may produce different valid outputs that are difficult to unify, making consensus-based arbitration challenging. To further verify this design choice, we added an additional experiment where the correction stage employs two agents. Specifically, the first agent produces an initial correction based on the model’s output, and this intermediate result is then passed to the second agent, which refines the final answer by incorporating both the original input and the first agent’s response. The results are as follow:
>
> | Detection | Localization | Correction | Detection Accuracy | Localization Accuracy | Correction ROUGE-1 | Correction BERTScore | Correction BLEURT | Average |
> |------------|---------------|-------------|--------------------|-----------------------|--------------------|----------------------|-------------------|----------|
> | 3 agent | 3 agent | 1 agent | 0.713 | 0.609 | 0.537 | 0.549 | 0.549 | 0.591 |
> | 3 agent | 3 agent | 2 agent | 0.713 | 0.609 | 0.367 | 0.367 | 0.374 | 0.486 |
>
> As shown, adding an extra correction agent decreases overall quality, confirming that a single-agent correction stage provides the most stable result within our framework.

---

> ### Author Response · Authors · 2025-11-23
> **Response to reviewer u4tR --- Part 4**
>
> > The parameter used in the KPCS weighting is described as tunable, but the paper does not specify which value was used for the reported results, impacting reproducibility and metric context. how sensitive is the model ranking to changes in this weighting.
>
> In our experiments, the weighting parameter alpha was set to 0.5. We have slightly revised the computation of KPCS by including samples with K() = 0, which makes the metric more comprehensive. Consequently, varying alpha affects how much the metric emphasizes important words versus overall semantics. The updated KPCS results are reported in Table 2. To assess the sensitivity of model rankings to this weighting, we further adjusted α to 0.2 and 0.8, and report the resulting rankings in the following Table (now included in Table 12):
>
> | Alpha | Gemini 2.0 Flash + MedGuards | GPT-4o- mini  + MedGuards | Doubao-1.5-thinking-pro + MedGuards | Deepseek-v3 + MedGuards | Gemini 2.5 Flash Lite + MedGuards |
> |:------:|:-----------------------------:|:------------------:|:----------------------------------:|:-----------------------:|:---------------------------------:|
> | 0.2 | 0.516 | 0.524 | 0.594 | 0.599 | 0.37 |
> | 0.5 | 0.472 | 0.461 | 0.554 | 0.556 | 0.385 |
> | 0.8 | 0.428 | 0.38 | 0.513 | 0.513 | 0.294 |
>
> As shown in the Table, varying Alpha does not lead to substantial changes in overall model ranking, but the margins between models shift because KPCS rebalances important-word vs. overall semantics as α changes
> A smaller α emphasizes overall semantics, while a larger α gives more weight to keywords.
>
> We also collected new human evaluation results (Normalized MOS Scores) from an additional clinician. The results are as follow:
>
> | Models                        | Normalized MOS Score |
> |-------------------------------|----------------------|
> | Gemini 2.0 Flash              | 0.425                |
> | Gemini 2.0 Flash + MedGuards  | 0.483                 |
> | GPT-4o-mini                   | 0.383                |
> | GPT-4o-mini + MedGuards       | 0.475                |
>
> The results indicate that α = 0.5 yields KPCS scores most consistent with human evaluation trends. For both Gemini 2.0 Flash and GPT-4o-mini, the model ranking under α = 0.5 best matches the human Normalized MOS results, while α = 0.2 or 0.8 tends to over or under emphasize differences.
>
> We kindly ask the reviewer to evaluate our submission in light of the above feedback and improvements made to our manuscript. Thank you for your time and effort.

---

> > ### Comment · Reviewer_u4tR · 2025-11-27
> >
> > Thanks for your comprehensive rebuttal, I have raised my scores accordingly.

---

### Meta-Review · Area_Chair_4DZD · 2026-01-09

**Summary:**

While the paper addresses an important and safety-critical problem in medical text generation and demonstrates a carefully engineered multi-agent pipeline with extensive experiments and rebuttal engagement, there are concerns about core novelty being limited, as the framework primarily recombines existing ideas (multi-agent self-consistency, CoT decomposition, confidence-based arbitration) without a clear advances beyond system integration.
There is also concern regarding a multi-agent architecture needing significant computational overhead and latency, which may limit its practical usefulness in in time-sensitive clinical settings. The work has merit, the authors are encouraged to continue their work and resubmit.

**Reviewer Concerns:**

The authors added meaningful ablations o

**Reviewer Scores:**

unchanged

---

### Decision · Program_Chairs · 2026-01-26

Reject